# Recovering Bandits

**Ciara Pike-Burke**[*]
Universitat Pompeu Fabra
Barcelona, Spain
c.pikeburke@gmail.com

**Steffen Grünewälder**
Lancaster University
Lancaster, UK
s.grunewalder@lancaster.ac.uk

## Abstract

We study the recovering bandits problem, a variant of the stochastic multi-armed bandit problem where the expected reward of each arm varies according to some unknown function of the time since the arm was last played. While being a natural extension of the classical bandit problem that arises in many real-world settings, this variation is accompanied by significant difficulties. In particular, methods need to plan ahead and estimate many more quantities than in the classical bandit setting. In this work, we explore the use of Gaussian processes to tackle the estimation and planing problem. We also discuss different regret definitions that let us quantify the performance of the methods. To improve computational efficiency of the methods, we provide an optimistic planning approximation. We complement these discussions with regret bounds and empirical studies.

## 1 Introduction

The multi-armed bandit problem [2, 29] is a sequential decision making problem, where, in each round $t$, we play an arm $J_t$ and receive a reward $Y_{t,J_t}$ generated from the unknown reward distribution of the arm. The aim is to maximize the total reward over $T$ rounds. Bandit algorithms have become ubiquitous in many settings such as web advertising and product recommendation. Consider, for example, suggesting items to a user on an internet shopping platform. This is modeled as a bandit problem where each product (or group of products) is an arm. Over time, the bandit algorithm will learn to suggest only good products. In particular, once the algorithm learns that a product (eg. a television) has high reward, it will continue to suggest it. However, if the user buys a television, the benefit of continuing to show it immediately diminishes, but may increase again as the television reaches the end of its lifetime. To improve customer experience (and profit), it would be beneficial for the recommendation algorithm to learn not to recommend the same product again immediately, but to wait an appropriate amount of time until the reward from that product has 'recovered'. Similarly, in film and TV recommendation, a user may wish to wait before re-watching their favorite film, or conversely, may want to continue watching a series but will lose interest in it if they haven't seen it recently. It would be advantageous for the recommendation algorithm to learn the different reward dynamics and suggest content based on the time since it was last seen. The recovering bandits framework presented here extends the stochastic bandit problem to capture these phenomena.

In the recovering bandits problem, the expected reward of each arm is given by an unknown function of the number of rounds since it was last played. In particular, for each arm $j$, there is a function $f_j(z)$ that specifies the expected reward from playing arm $j$ when it has not been played for $z$ rounds. We take a Bayesian approach and assume that the $f_j$'s are sampled from a Gaussian process (GP) (see Figure 1a). Using GPs allows us to capture a wide variety of functions and deal appropriately with uncertainty. For any round $t$, let $Z_{j,t}$ be the number of rounds since arm $j$ was last played. This changes for both the played arm (it resets to 0) and also for the unplayed arms (it increases by 1) in

---

[*]This work was carried out while CPB was a PhD student at STOR-i, Lancaster University, UK.

every round. Thus, the expected reward of every arm changes in every round, and this change depends on whether it was played. This problem is therefore related to both restless and rested bandits [30].

In recovering bandits, the reward of each arm depends on the entire sequence of past actions. This means that, even if the recovery functions were known, selecting the best sequence of $T$ arms is intractable (since, in particular, an MDP representation would be unacceptably large). One alternative is to select the action that maximizes the *instantaneous* reward, without considering future decisions. This is still quite a challenge compared to the $K$-armed bandit problem, as instead of just learning the reward of each arm, we must learn recovery functions. In some cases, maximizing the instantaneous reward is not optimal. In particular, using knowledge of the reward dynamics, it is often possible to find a sequence of arms whose total reward is greater than that gained by playing the instantaneous greedy arms. Thus, our interest lies in selecting sequences of arms to maximize the reward.

In this work, we present and analyze an Upper Confidence Bound (UCB) [2] and Thompson Sampling [29] algorithm for recovering bandits. By exploiting properties of Gaussian processes, both of these accurately estimate the recovery functions and uncertainty, and use these to look ahead and select sequences of actions. This leads to strong theoretical and empirical performance. The paper proceeds as follows. In Section 2 we discuss related work. We formally define our problem in Section 3 and the regret in Section 4. In Section 5, we present our algorithms and bound their regret. We use optimistic planning in Section 6 to improve computational complexity and show empirical results in Section 7 before concluding.

## 2 Related Work

In the restless bandits problem, the reward distribution of any arm changes at any time, regardless of whether it is played. This problem has been studied by [30, 27, 10, 23, 4] and others. In the rested bandits problem, the reward distribution of an arm only changes when it is played. [17, 8, 6, 12, 26] study rested bandits problems with rewards that vary mainly with the number of plays of an arm.

In recovering bandits, the rewards depend on the time since the arm was last played. [14] consider concave and increasing recovery functions and [31] study recommendation algorithms with known step recovery functions. The closest work to ours is [18] where the expected reward of each arm depends on a state (which could be the time since the arm was played) via a parametric function. They use maximum likelihood estimation (although there are no guarantees of convergence) in a KL-UCB algorithm [7]. The expected frequentist regret of their algorithm is $O(\sum_j \log(T)/\delta_j^2)$ where $\delta_j$ depends on the random number of plays of arm $j$ and the minimum difference in the rewards of any arms at any time (which can be very small). By the standard worst case analysis, the frequentist problem independent regret is $O^*(T^{2/3}K^{1/3})$, where we use the notation $O^*$ to suppress log factors. Our algorithms achieve $O^*(\sqrt{KT})$ Bayesian regret and require less knowledge of the recovery functions. [18] also provide an algorithm with no theoretical guarantees but improved experimental performance. In Section 7, we show that our algorithms outperform this algorithm experimentally.

In Gaussian process bandits, there is a function, $f$, sampled from a GP and the aim is to minimize the (Bayesian) regret with respect to the maximum of $f$. The GP-UCB algorithm of [28] has Bayesian regret $O^*(\sqrt{T\gamma_T})$ where $\gamma_T$ is the maximal information gain (see Section 5). By [25], Thompson sampling has the same Bayesian regret. [5] consider GP bandits with a slowly drifting reward function and [16] study contextual GP bandits. These contexts and drifts do not depend on previous actions.

It is important to note that all of the above approaches only look at instantaneous regret whereas in recovering bandits, it is more appropriate to consider lookahead regret (see Section 4). We will also consider Bayesian regret. Many naive approaches will not perform well in this problem. For example, treating each $(j, z)$ combination as an arm and using UCB [2] with $K|\mathcal{Z}|$ arms leads to regret $O^*(\sqrt{KT|\mathcal{Z}|})$ (see Appendix F). Our algorithms exhibit only $\sqrt{\log |\mathcal{Z}|}$ dependence on $|\mathcal{Z}|$. Adversarial bandit algorithms will not do well in this setting either since they aim to minimize the regret with respect to the best constant arm, which is clearly suboptimal in recovering bandits.

## 3 Problem Definition

We have $K$ independent arms and play for $T$ rounds ($T$ is not known). For each arm $1 \leq j \leq K$ and round $1 \leq t \leq T$, denote by $Z_{j,t}$ the number of rounds since arm $j$ was last played, where

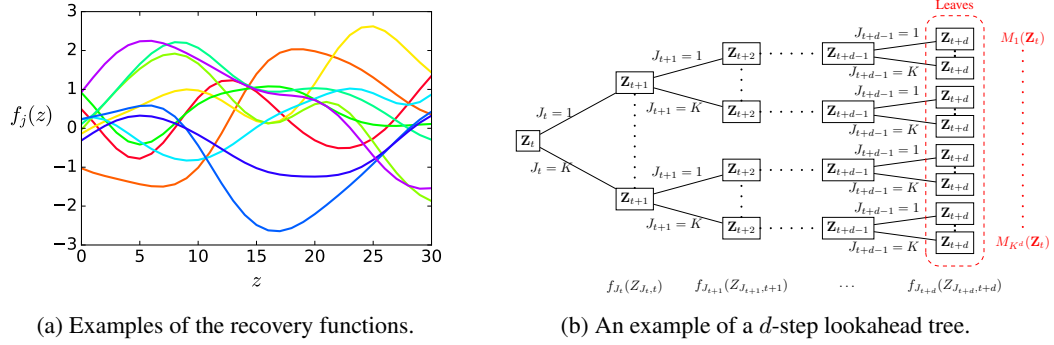

(a) Examples of the recovery functions.  (b) An example of a $d$-step lookahead tree.

Figure 1: Illustration of recovery functions and lookahead trees.

$Z_{j,t} \in \mathcal{Z} = \{0, \ldots, z_{\max}\}$ for finite $z_{\max} \in \mathbb{N}$. If we play arm $J_t$ at time $t$ then, at time $t+1$,

$$Z_{j,t+1} = \begin{cases} 0 & \text{if } J_t = j, \\ \min\{z_{\max}, Z_{j,t}+1\} & \text{if } J_t \neq j. \end{cases} \tag{1}$$

Hence, if arm $j$ has not been played for more than $z_{\max}$ steps, $Z_{j,t}$ will stay at $z_{\max}$. The $Z_{j,t}$ are random variables since they depend on our past actions. We will assume that $T \geq K|\mathcal{Z}|$.

The expected reward for arm $j$ is modeled by an (unknown) recovery function, $f_j$. We assume that the $f_j$'s are sampled independently from a Gaussian processes with mean 0 and known kernel (see Figure 1a). Let $\mathbf{Z}_t = (Z_{1,t}, \ldots, Z_{K,t})$ be the vector of covariates for each arm at time $t$. At round $t$, we observe $\mathbf{Z}_t$ and use this and past observations to select an arm $J_t$ to play. We then receive a noisy observation $Y_{J_t,t} = f_{J_t}(Z_{J_t,t}) + \epsilon_t$ where $\epsilon_t$ are i.i.d. $\mathcal{N}(0, \sigma^2)$ random variables and $\sigma$ is known.

[24] give an introduction to Gaussian Processes (GP). A Gaussian process gives a distribution over functions, when for every finite set $z_1, \ldots, z_N$ of covariates, the joint distribution of $f(z_1), \ldots, f(z_N)$ is Gaussian. A GP is defined by its mean function, $\mu(z) = \mathbb{E}[f(z)]$, and kernel function, $k(z, z') = \mathbb{E}[(f(z) - \mu(z))(f(z') - \mu(z'))]$. If we place a GP prior on $f$ and observe $\mathbf{Y}_N = (Y_1, \ldots, Y_N)^T$ at $\mathbf{z}_N = (z_1, \ldots, z_N)^T$ where $Y_n = f(z_n) + \epsilon_n$ for $\epsilon_n$ iid $\mathcal{N}(0, \sigma^2)$ noise, then the posterior distribution after $N$ observations is $\mathcal{GP}(\mu(z; N), k(z, z'; N))$. Here for $\mathbf{k}_N(z) = (k(z_1, z), \ldots, k(z_N, z))^T$ and positive semi-definite kernel matrix $\mathbf{K}_N = [k(z_i, z_j)]_{i,j=1}^N$, the posterior mean and covariance are,

$$\mu(z; N) = \mathbf{k}_N(z)^T(\mathbf{K}_N + \sigma^2\mathbf{I})^{-1}\mathbf{y}_N, \quad k(z, z; N) = k(z, z') - \mathbf{k}_N(z)^T(\mathbf{K}_N + \sigma^2\mathbf{I})^{-1}\mathbf{k}_N(z'),$$

so $\sigma^2(z; N) = k(z, z; N)$. For $z \in \mathcal{Z}$, the posterior distribution of $f(z)$ is $\mathcal{N}(\mu(z; N), \sigma^2(z; N))$. We consider the posterior distribution of $f_j$ for each arm at every round, when it has been played some (random) number of times. For each arm $j$, denote the posterior mean and variance of $f_j$ at $z$ after $n$ plays of the arm by $\mu_j(z; n)$ and $\sigma_j^2(z; n)$. Let $N_j(t)$ be the (random) number of times arm $j$ has been played up to time $t$. We denote the posterior mean and variance of arm $j$ at round $t$ by,

$$\mu_t(j) = \mu_j(Z_{j,t}; N_j(t-1)), \qquad \text{and} \qquad \sigma_t^2(j) = \sigma_j^2(Z_{j,t}; N_j(t-1)).$$

## 4  Defining the Regret

The regret is commonly used to measure the performance of an algorithm and is defined as the difference in the cumulative expected reward of an algorithm and an oracle. We will use the Bayesian regret, where the expectation is taken over the recovery curves and the actions. In recovering bandits, there are various choices for the oracle. We discuss some of these here.

**Full Horizon Regret.**  One candidate for the oracle is the deterministic policy which knows the recovery functions and $T$, and using this selects the best sequence of $T$ arms. This policy can be horizon dependent. Anytime algorithms, which are horizon independent, lead to policies that are stationary and do not change over time. In various settings, these stationary deterministic policies achieve the best possible regret [22]. In the following, we focus on the stationary deterministic (SD) oracle. Note that it is computationally intractable to calculate this oracle in all but the easiest problems. This can be seen by formulating the problem as an MDP, with natural state-space of size $K^{|\mathcal{Z}|}$. Techniques such as dynamic programming cannot be used unless $K$ and $|\mathcal{Z}|$ are very small.

**Instantaneous Regret.** Another candidate for the oracle is the policy which in each round $t$, greedily plays the arm with the highest immediate reward at $\mathbf{Z}_t$. These $\mathbf{Z}_t$ depend on the previous actions of the oracle. Consider a policy which plays this oracle up to time $s-1$, then selects a different action at time $s$, and continues to play greedily. The cumulative reward of this policy could be vastly different to that of the oracle since they may have very different $\mathbf{Z}$ values. Therefore, defining regret in relation to this oracle may penalize us severely for early mistakes. Instead, one can define the regret of a policy $\pi$ with respect to an oracle which selects the best arm *at the $\mathbf{Z}_t$'s generated by $\pi$*. We call this the *instantaneous regret*. This regret is commonly used in restless bandits and in [18].

$d$**-step Lookahead Regret.** A policy with low instantaneous regret may miss out on additional reward by not considering the impact of its actions on future $\mathbf{Z}_t$'s. Looking ahead and considering the evolution of the $Z_{j,t}$'s can lead to choosing sequences of arms which are collectively better than individual greedy arms. For example, if two arms $j_1, j_2$ have similar $f_j(Z_{j,t})$ but the reward of $j_1$ doubles if we do not play it, while the reward of $j_2$ stays the same, it is better to play $j_2$ then $j_1$. We will consider oracles which take the $\mathbf{Z}_t$ generated by our algorithm and select the best sequence of $d \geq 1$ arms. We call this regret the $d$-*step lookahead regret* and will use this throughout the paper.

To define this regret, we use decision trees. Nodes are $\mathbf{Z}$ values and edges represent playing arms and updating $\mathbf{Z}$ (see Figure 1b). Each sequence of $d$ arms is a leaf of the tree. Let $\mathcal{L}_d(\mathbf{Z})$ be the set of leaves of a $d$-step lookahead tree with root $\mathbf{Z}$. For any $i \in \mathcal{L}_d(\mathbf{Z})$, denote by $M_i(\mathbf{Z})$ the expected reward at that leaf, that is the sum of the $f_j$'s along the path to $i$ at the relevant $Z_j$'s (see Section 5). The $d$-step lookahead oracle selects the leaf with highest $M_i(\mathbf{Z}_t)$ from a given root node $\mathbf{Z}_t$, denote this value by $M^*(\mathbf{Z}_t)$. This leaf is the best sequence of $d$ arms from $\mathbf{Z}_t$. If we select leaf $I_t$ at time $t$, we play the arms to $I_t$ for $d$ steps, so select a leaf every $d$ rounds. The $d$-step lookahead regret is,

$$\mathbb{E}[\mathfrak{R}_T^{(d)}] = \sum_{h=0}^{\lfloor T/d \rfloor} \mathbb{E}\left[ M^*(\mathbf{Z}_{hd+1}) - M_{I_{hd+1}}(\mathbf{Z}_{hd+1}) \right],$$

with expectation over $I_{hd+1}$ and $f_j$. If $d = T$ or $d = 1$, we get the full horizon or instantaneous regret. We study the single play regret, $\mathbb{E}[\mathfrak{R}_T^{(d,s)}]$, where arms can only be played once in a $d$-step lookahead, and the multiple play regret, $\mathbb{E}[\mathfrak{R}_T^{(d,m)}]$, which allows multiple plays of an arm in a lookahead. This regret is related to that in episodic reinforcement learning (ERL) [15, 21, 3]. A key difference is that in ERL, the initial state is reset or re-sampled every $d$ steps independent of the actions taken. Note that the $d$-step lookahead regret can be calculated for any policy, regardless of whether the policy is designed to look ahead and select sequences of $d$ actions.

For large $d$, the total reward from the optimal $d$-step lookahead policy will be similar to that of the optimal full horizon stationary deterministic policy. Let $V_T(\pi)$ be the total reward of policy $\pi$ up to horizon $T$ and note that the optimal SD policy will be periodic by Lemma 16 (Appendix D). Then,

**Proposition 1** *Let $p^*$ be the period of the optimal SD policy $\pi^*$. For any $l = 1, \ldots, \lfloor \frac{T - z_{\max}}{p^*} \rfloor$, the optimal $(z_{\max} + lp^*)$-lookahead policy, $\pi_l^*$, satisfies, $V_T(\pi_l^*) \geq \left(1 - \frac{(l+1)p^* + z_{\max}}{T + p^*}\right) \frac{lp^*}{lp^* + z_{\max}} V_T(\pi^*)$.*

Hence, any algorithm with low $(z_{\max} + lp^*)$-step lookahead regret will also have high total reward. In practice, we may not know the periodicity of $\pi^*$. Moreover, if $p^*$ is too large, then looking $(z_{\max} + lp^*)$ steps ahead may be computationally challenging, and prohibit learning. Hence, we may wish to consider smaller values of $d$. One option is to look far enough ahead that we consider a local maximum of each recovery function. For a GP kernel with lengthscale $l$ (e.g. squared exponential or Matérn), this requires looking $2l$ steps ahead [20, 24]. This should still give large reward while being computationally more efficient and allowing for learning.

## 5 Gaussian Processes for Recovering Bandits

In Algorithm 1 we present a UCB ($d$RGP-UCB) and Thompson Sampling ($d$RGP-TS) algorithm for the $d$-step lookahead recovering bandits problem, for both the single and multiple play case. Our algorithms use Gaussian processes to model the recovery curves, allowing for efficient estimation and facilitating the lookahead. For each arm $j$ we place a GP prior on $f_j$ and initialize $Z_{j,1}$ (often this initial value is known, otherwise we set it to 0). Every $d$ steps we construct the $d$-step lookahead

---
**Algorithm 1** $d$-step lookahead UCB and Thompson Sampling
---
**Input:** $\alpha_t$ from (3) (for UCB).
**Initialization:** Define $\mathcal{T}_d = \{1, d+1, 2d+1, \dots\}$. For all arms $j \in A$, set $Z_{j,1} = 0$ (optional).
**for** $t \in \mathcal{T}_d$ **do**
    If $t \geq T$ **break**. Else, construct the $d$-step lookahead tree. Then,

    If UCB:                                  If TS: (i) $\forall j \in A$, sample $\tilde{f}_j$ from the posterior at $\mathbf{Z}_{j,t}^{(d)}$.

$$I_t = \underset{i \in \mathcal{L}_d(\mathbf{Z}_t)}{\operatorname{argmax}} \left\{ \eta_t(i) + \alpha_t \varsigma_t(i) \right\}, \qquad \begin{array}{l} \text{(ii) } \forall i \in \mathcal{L}_d(\mathbf{Z}_t), \tilde{\eta}_t(i) = \sum_{l=0}^{d-1} \tilde{f}_{J_{t+\ell}}(Z_{J_{t+\ell}, t+\ell}) \\ \text{(iii) } I_t = \operatorname{argmax}_{i \in \mathcal{L}_d(\mathbf{Z}_t)} \{\tilde{\eta}_t(i)\} \end{array}$$

    **for** $\ell = 0, \dots, d-1$ **do**
        Play $\ell$th arm to $I_t$, $J_\ell$, and get reward $Y_{J_\ell, t+\ell}$.
        Set $Z_{J_\ell, t+\ell+1} = 0$. For all $j \neq J_\ell$, set $Z_{j,t+\ell+1} = \min\{Z_{j,t+\ell} + 1, z_{\max}\}$.
    **end for**
    Update the posterior distributions of the played arms.
**end for**
---

tree as in Figure 1b. At time $t$, we select a sequence of arms by choosing a leaf $I_t$ of the tree with root $\mathbf{Z}_t$. For a leaf $i \in \mathcal{L}_d(\mathbf{Z}_t)$, let $\{J_{t+\ell}\}_{\ell=0}^{d-1}$ and $\{Z_{J_{t+\ell}, t+\ell}\}_{\ell=0}^{d-1}$ be the sequences of arms and $z$ values (which are updated using (1)) on the path to leaf $i$. Then define the total reward at $i$ as,

$$M_i(\mathbf{Z}_t) = \sum_{\ell=0}^{d-1} f_{J_{t+\ell}}(Z_{J_{t+\ell}, t+\ell}).$$

Since the posterior distribution of $f_j(z)$ is Gaussian, the posterior distribution of the leaves of the lookahead tree will also be Gaussian. In particular, $\forall i \in \mathcal{L}_d(\mathbf{Z}_t)$, $M_i(\mathbf{Z}_t) \sim \mathcal{N}(\eta_t(i), \varsigma_t^2(i))$ where,

$$\eta_t(i) = \sum_{\ell=0}^{d-1} \mu_t(J_{t+\ell}), \quad \varsigma_t^2(i) = \sum_{\ell,q=0}^{d-1} \operatorname{cov}_t(f_{J_{t+\ell}}(Z_{J_{t+\ell}, t+\ell}), f_{J_{t+q}}(Z_{J_{t+q}, t+q})), \text{ and,} \quad (2)$$

$\operatorname{cov}_t(f_{J_{t+\ell}}(Z_{J_{t+\ell}, t+\ell}), f_{J_{t+q}}(Z_{J_{t+q}, t+q})) = \mathbb{I}\{J_{t+\ell} = J_{t+q}\} k_{J_{t+\ell}}(Z_{J_{t+\ell}, t+\ell}, Z_{J_{t+q}, t+q}; N_{J_{t+\ell}}(t))$.
Hence, using GPs enables us to accurately estimate the reward and uncertainty at the leaves.

For $d$RGP-UCB, we construct upper confidence bounds on each $M_i(\mathbf{Z}_t)$ using Gaussianity. We then select the leaf $I_t$ with largest upper confidence bound at time $t$. That is,

$$I_t = \underset{1 \leq i \leq K^d}{\operatorname{argmax}} \{\eta_t(i) + \alpha_t \varsigma_t(i)\} \qquad \text{where} \qquad \alpha_t = \sqrt{2 \log((K|\mathcal{Z}|)^d (t+d-1)^2)}. \quad (3)$$

In $d$RGP-TS, we select a sequence of $d$ arms by sampling the recovery function of each arm $j$ at $\mathbf{Z}_{j,t}^{(d)} = (Z_{j,t}, \dots, Z_{j,t} + d - 1, 0, \dots, d - 1)^T$ and then calculating the 'reward' of each node using these sampled values. Denote the sampled reward of node $i$ by $\tilde{\eta}_t(i)$. We choose the leaf $I_t$ with highest $\tilde{\eta}_t(i)$. In both $d$RGP-UCB and $d$RGP-TS, by the lookahead property, we will only play an arm at a large $Z_{j,t}$ value if it has high reward, or high uncertainty there. We play the sequence of $d$ arms indicated by $I_t$ over the next $d$ rounds. We then update the posteriors and repeat the process.

We analyze the regret in the single and multiple play cases separately. Studying the single play case first allows us to gain more insights about the difficulty of the problem. Indeed, from our analysis we observe that the multiple play case is more difficult since we may loose information from not updating the posterior between plays of the same arm. All proofs are in the appendix. The regret of our algorithms will depend on the GP kernel through the maximal information gain. For a set $\mathcal{S}$ of covariates and observations $Y_{\mathcal{S}} = [f(z) + \epsilon_z]_{z \in \mathcal{S}}$, we define the *information gain*, $\mathcal{I}(Y_{\mathcal{S}}; f) = H(Y_{\mathcal{S}}) - H(Y_{\mathcal{S}}|f)$ where $H(\cdot)$ is the entropy. As in [28], we consider the *maximal information gain* from $N$ samples, $\gamma_N$. If $z_t \in \mathcal{S}$ is played at time $t$, then,

$$\mathcal{I}(Y_{\mathcal{S}}, f) = \frac{1}{2} \sum_{t=1}^{|\mathcal{S}|} \log(1 + \sigma^{-2} \sigma^2(z_t; t-1)), \qquad \text{and,} \qquad \gamma_N = \max_{\mathcal{S} \subset \mathcal{Z}^N : |\mathcal{S}| = N} \mathcal{I}(Y_{\mathcal{S}}; f). \quad (4)$$

Theorem 5 of [28] gives bounds on $\gamma_T$ for some kernels. We apply these results using the fact that the dimension, $D$, of the input space is 1. For any lengthscale, $\gamma_T = O(\log(T))$ for linear kernels, $\gamma_T = O(\log^2(T))$ for squared exponential kernels, and $\gamma_T = O(T^{2/(2\nu+2)} \log(T))$ for Matérn($\nu$).

## 5.1 Single Play Lookahead

In the single play case, each am can only be played once in the $d$-step lookahead. This simplifies the variance in (2) since the arms are independent. For any leaf $i$ corresponding to playing arms $J_t, \ldots, J_{t+d-1}$ (at the corresponding $z$ values), $\varsigma_t^2(i) = \sum_{\ell=0}^{d-1} \sigma_t^2(J_{t+\ell})$. This involves the posterior variances at time $t$. However, as we cannot repeat arms, if we play arm $j$ at time $t+\ell$ for $0 \le \ell \le d-1$, it cannot have been played since time $t$, so its posterior is unchanged. Using this and (4), we relate the variance of $M_{I_t}(\mathbf{Z}_t)$ to the information gain about the $f_j$'s. We get the following regret bounds.

**Theorem 2** *The $d$-step single play lookahead regret of dRGP-UCB satisfies,*
$$\mathbb{E}[\mathfrak{R}_T^{(d,s)}] \le O(\sqrt{KT\gamma_T \log(TK|\mathcal{Z}|)}).$$

**Theorem 3** *The $d$-step single play lookahead regret of dRGP-TS satisfies,*
$$\mathbb{E}[\mathfrak{R}_T^{(d,s)}] \le O(\sqrt{KT\gamma_T \log(TK|\mathcal{Z}|)}).$$

## 5.2 Multiple Play Lookahead

When arms can be played multiple times in the $d$-step lookahead, the problem is more difficult since we cannot use feedback from plays within the same lookahead to inform decisions. It is also harder to relate $\varsigma_t^2(I_t)$ to the information gain about each $f_j$. In particular, $\varsigma_t^2(I_t)$ contains covariance terms and is defined using the posteriors at time $t$. On the other hand, $\gamma_T$ is defined in terms of the posterior variances when each arm is played. These may be different to those at time $t$ if an arm is played multiple times in the lookahead. However, using the fact that the posterior covariance matrix is positive semi-definite, $2k_j(z_1, z_2; n) \le \sigma_j^2(z_1; n) + \sigma_j^2(z_2; n)$, so we can bound $\varsigma_t^2(I_t) \le 3 \sum_{\ell=0}^{d-1} \sigma_t^2(J_{t+\ell})$. Then, the change in the posterior variance of a repeated arm can be bounded by the following lemma.

**Lemma 4** *For any $z \in \mathcal{Z}$, arm $j$ and $n \in \mathbb{N}, n \ge 1$, let $Z_j^{(n)}$ be the $z$ value at the $n$th play of arm $j$. Then, $\sigma_j^2(z; n-1) - \sigma_j^2(z; n) \le \sigma^{-2}\sigma_j^2(Z_j^{(n)}; n-1)$.*

We get the following regret bounds for $d$RGP-UCB and $d$RGP-TS. Due to not updating the posterior between repeated plays of an arm, they increase by a factor of $\sqrt{d}$ compared to the single play case. Thus, although by Proposition 1 larger $d$ leads to higher reward, it makes the learning problem harder.

**Theorem 5** *The $d$-step multiple play lookahead regret of dRGP-UCB satisfies,*
$$\mathbb{E}[\mathfrak{R}_T^{(d,m)}] \le O\left(\sqrt{KT\gamma_T \log((K|\mathcal{Z}|)^d T)}\right).$$

**Theorem 6** *The $d$-step multiple play lookahead regret of dRGP-TS satisfies,*
$$\mathbb{E}[\mathfrak{R}_T^{(d,m)}] \le O\left(\sqrt{KT\gamma_T \log((K|\mathcal{Z}|)^d T)}\right).$$

## 5.3 Instantaneous Algorithm

If we set $d = 1$ in Algorithm 1, we obtain algorithms for minimizing the instantaneous regret. In this case, $\mathcal{T} = \{1, \ldots, T\}$ and there are $K$ leaves of the 1-step lookahead tree, so each $M_i(\mathbf{Z}_t)$ corresponds to one arm. One arm is selected and played each time step so $\eta_t(i) = \mu_t(j)$, $\varsigma_t^2(i) = \sigma_t^2(j)$ for some $j$. For the UCB, we define $\alpha_t$ as in (3) with $d = 1$. We get the following regret,

**Corollary 7** *The instantaneous regret of 1RGP-UCB and 1RGP-TS up to horizon $T$ satisfy*
$$\mathbb{E}[\mathfrak{R}_T^{(1)}] \le O(\sqrt{KT\gamma_T \log(TK|\mathcal{Z}|)}).$$

The instantaneous regret of both algorithms is $O^*(\sqrt{KT\gamma_T})$. Hence, we reduced the dependency on $|\mathcal{Z}|$ from $\sqrt{|\mathcal{Z}|}$ to $\sqrt{\log|\mathcal{Z}|}$ compared to a naive application of UCB (see Appendix F). The single play lookahead regret is of the same order as this instantaneous regret. This shows that, in the single play case, since we still update the posterior after every play of an arm, we do not loose any information by looking ahead.

# 6 Improving Computational Efficiency via Optimistic Planning

For large values of $K$ and $d$, Algorithm 1 may not be computationally efficient since it searches $K^d$ leaves. We can improve this by optimistic planning [13, 19]. This was developed by [13] for deterministic MDPs with discount factors and rewards in $[0, 1]$. We adapt this to undiscounted rewards in $[\min_{j,z} \tilde{f}_j(z), \max_{j,z} \tilde{f}_j(z)]$. We focus on this in the multiple play Thompson sampling algorithm.

As in Algorithm 1, at time $t$, we sample $\tilde{f}_j(z)$ from the posterior of $f_j$ at $\mathbf{Z}_{j,t}^{(d)} = (Z_{j,t}, \dots, Z_{j,t} + d, 0, \dots d)^T$ for all arms $j$. Then, instead of searching the entire tree to find the leaf with largest total $\tilde{f}_j(z)$, we use optimistic planning (OP) to iteratively build the tree. We start from an initial tree of one node, $i_0 = \mathbf{Z}_t$. At step $n$ of the OP procedure, let $\mathcal{T}_n$ be the expanded tree and let $\mathcal{S}_n$ be the nodes not in $\mathcal{T}_n$ but whose parents are in $\mathcal{T}_n$. We select a node in $\mathcal{S}_n$ and move it to $\mathcal{T}_n$, adding its children to $\mathcal{S}_n$. If we select a node $i_n$ of depth $d$, we stop and output node $i_n$. Otherwise we continue until $n = N$ for a predefined budget $N$. Let $d_N$ be the maximal depth of nodes in $\mathcal{T}_N$. We output the node at depth $d_N$ with largest upper bound on the value of its continuation (i.e. with largest $b_N(i)$ in (5)).

Nodes are selected using upper bounds on the total value of a continuation of the path to the node. For node $i \in \mathcal{S}_n \cup \mathcal{T}_n$, let $u(i)$ be the sum of the $\tilde{f}_j(z)$'s on the path to $i$, and $l(i)$ the depth of $i$. The value, $v(i)$, of node $i$ is the reward to $i$, $u(i)$, plus the maximal reward of a path from $i$ to depth $d$. We upper bound $v(i)$ by,

$$b_n(i) = u(i) + \Psi(\mathbf{z}(i), d - l(i)) \qquad \text{for } i \in \mathcal{S}_n \cup \mathcal{T}_n. \tag{5}$$

where $\mathbf{z}(i)$ is the vector of $z_j$'s at node $i$, and the function $\Psi(\mathbf{z}(i), d - l(i))$ is an upper bound on the maximal reward from node $i$ to a leaf. Let $g_j(z, l) = \max\{\tilde{f}_j(z), \dots, \tilde{f}_j(z+l), \tilde{f}_j(0), \dots, \tilde{f}_j(l)\}$ be the maximal reward that can be gained from playing arm $j$ in the next $1 \leq l \leq d$ steps from $z \in \mathbf{Z}_{j,t}^{(d)}$. Then, $\Psi(\mathbf{z}(i), d - l(i)) = (d - l(i)) \max_{1 \leq j \leq K} g_j(z_j(i), d - l(i))$, and $\Psi(\mathbf{z}(i), 0) = 0$ for any $\mathbf{z}(i)$.

We can often bound the error from this procedure. Let $v^* = \max_{i \in \mathcal{L}_d(\mathbf{Z})} v(i)$ be the value of the maximal node. A node $i$ is $\epsilon$-optimal if $v^* - v(i) \leq \epsilon$, and let $p_l(\epsilon)$ be the proportion of $\epsilon$-optimal nodes at depth $l$. Let $\Delta = \max_{j,z} \tilde{f}_j(z) - \min\{\min_{j,z} \tilde{f}_j(z), 0\}$ and define $\Psi^*(l) = \max_{\mathbf{z} \in \mathcal{Z}} \Psi(\mathbf{z}, l)$ for any $l = 0, 1, \dots, d$. We get the following bound (whose proof is in Appendix E).

**Proposition 8** *For the optimistic planning procedure with budget $N$, if the procedure stops at step $n < N$ because a node $i_n$ of depth $d$ is selected, then $v^* - v(i_n) = 0$. Otherwise, if there exist $\lambda \in (\frac{1}{K}, 1]$ and $1 \leq d_0 \leq d$ such that $\forall l \geq d_0$, $p_l((d-l)\Delta) \leq \lambda^l$, then for $N > n_0 = \frac{K^{d_0+1}-1}{K-1}$,*

$$v^* - v(i_N) \leq \left( d - \frac{\log(N - n_0)}{\log(\lambda K)} - \frac{\log(\lambda K - 1)}{\log(\lambda K)} + 1 \right) \Delta. \tag{6}$$

Hence, if we stop the procedure at $n < N$, the node $i_n$ of depth $d$ we return will be optimal. In many cases, especially when the proportion of $\epsilon$-optimal nodes, $\lambda$, is small, this will occur. Otherwise, the error will depend on $\lambda$, and the budget, $N$. By (6), for $N \approx (\lambda K)^d$, the error will be near zero.

# 7 Experimental Results

We tested our algorithms in experiments with $z_{\max} = 30$, noise standard deviation $\sigma = 0.1$, and horizon $T = 1000$. We used GPy [11] to fit the GPs. We first aimed to check that our algorithms were playing arms at good $z$ values (i.e. play arm $j$ when $f_j(z)$ is high). We used $K = 10$ and sampled the recovery functions from a squared exponential kernel and ran the algorithms once. Figure 2 shows that, for lengthscale $l = 2$, 1RGP-UCB and 3RGP-UCB accurately estimate the recovery functions and learn to play each arm in the regions of $\mathcal{Z}$ where the reward is high. Although, as expected, 3RGP-UCB has more samples on the peaks, it is reassuring that the instantaneous algorithm also selects good $z$'s. The same is true for $d$RGP-TS and different values of $d$ and $l$ (see Appendix G.1).

Next, we tested the performance of using optimistic planning (OP) in $d$RGP-TS. We averaged all results over 100 replications and used a squared exponential kernel with $l = 4$. In the first case, $K = 10$ and $d = 4$, so direct tree search may have been possible. Figure 3a shows that, when $N$, the budget of policies the OP procedure can evaluate per lookahead, increases above 500, the total

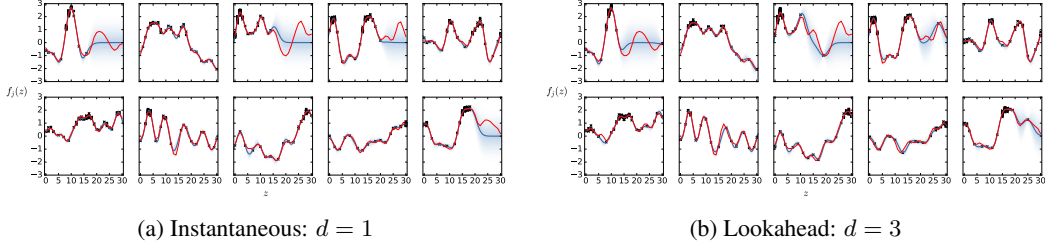

|                        |                        |
| ---------------------- | ---------------------- |
| (a) Instantaneous: $d = 1$ | (b) Lookahead: $d = 3$ |

Figure 2: The posterior mean (blue) of RGP-UCB with density shaded in blue for a squared exponential kernel ($l = 2$). The true recovery curve is in red and the crosses are the observed samples.

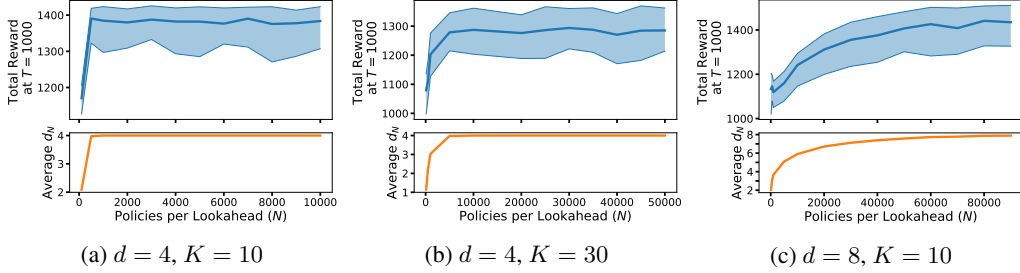

|                        |                        |                        |
| ---------------------- | ---------------------- | ---------------------- |
| (a) $d = 4$, $K = 10$  | (b) $d = 4$, $K = 30$  | (c) $d = 8$, $K = 10$  |

Figure 3: The total reward and final depth of the lookahead tree, $d_N$, as the budget, $N$, increases.

reward plateaus and the average depth of the returned policy is 4. By Proposition 8, this means that we found an optimal leaf of the lookahead tree while evaluating significantly fewer policies. We then increased the number of arms to $K = 30$. Here, searching the whole lookahead tree would be computationally challenging. Figure 3b shows that we found the optimal policy after about 5,000 policies (since here $d_N = d$), which is less than $0.1\%$ of the total number of policies. In Figure 3c, we increased the depth of the lookahead to $d = 8$. In this case, we had to search more policies to find optimal leaves. However, this was still less than $0.1\%$ of the total number of policies. From Figure 3c, we also see that when $d_N < d$, increasing $N$ leads to higher total reward.

Lastly, we compared our algorithms to RogueUCB-Tuned [18] and UCB-Z, the basic UCB algorithm with $K|Z|$ arms (see Appendix F), in two parametric settings. Details of the implementation of RogueUCB-Tuned are given in Appendix G.2. As in [18], we only considered $d = 1$. We used squared exponential kernels in 1RGP-UCB and 1RGP-TS with lengthscale $l = 5$ (results for other lengthscales are in Appendix G.3). The recovery functions were logistic, $f(z) = \theta_0(1 + \exp\{-\theta_1(z - \theta_2)\})^{-1}$ which increases in $z$, and modified gamma, $f(z) = \theta_0 C \exp\{-\theta_1 z\} z^{\theta_2}$ (with normalizer $C$), which increases until a point and then decreases. The values of $\theta$ were sampled uniformly and are in Appendix G.3. We averaged the results over 500 replications. The cumulative regret (and confidence regions) is shown in Figure 4 and the cumulative reward (and confidence bounds) in Table 1. Our algorithms achieve lower regret and higher reward than RogueUCB-Tuned. UCB-Z does badly since the time required to play each (arm,$z$) pair during initialization is large.

### 7.1 Practical Considerations

There are several issues to consider when applying our algorithms in a practical recommendation scenario. The first is the choice of $z_{\max}$. Throughout, we assumed that this is known and constant across arms. Our work can be easily extended to the case where there is a different value, $z_{\max,j}$, for each arm $j$, by defining $z_{\max} = \max_j z_{\max,j}$ and extending $f_j$ to $z_{\max}$ by setting $f_j(z) = f_j(z_{\max,j})$ for all $z = z_{\max,j} + 1, \ldots, z_{\max}$. A similar approach can be used if we only know an upper bound on $z_{\max}$. Additionally, in practice, the recovery curves may not be sampled from Gaussian processes, or the kernels may not be known. As demonstrated experimentally, our algorithms can still perform well in this case. Indeed, kernels can be chosen to (approximately) represent a wide variety of recovery curves, ranging from uncorrelated rewards to constant functions. In practice, we can also use adaptive algorithms for selecting a kernel function out of a large class of kernel functions (see e.g. Chapter 5 of [24] for details).

Table 1: Total reward at $T = 1000$ for single step experiments with parametric functions

| Setting | 1RGP-UCB ($l = 5$) | 1RGP-TS ($l = 5$) | RogueUCB-Tuned | UCB-Z |
|---|---|---|---|---|
| Logistic | 461.7 (454.3,468.9) | 462.6 (455.7,469.3) | 446.2 (438.2,453.5) | 242.6 (229.6,256.0) |
| Gamma | 145.6 (139.6, 151.7) | 156.5 (149.6,163.0) | 132.7 (111.0,144.5) | 116.8 (108.4,125.5) |

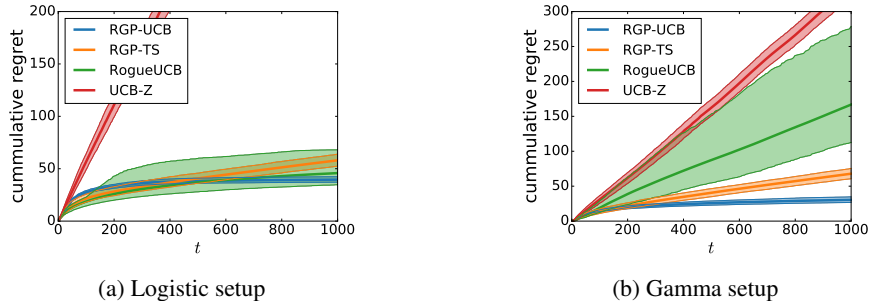

(a) Logistic setup

(b) Gamma setup

Figure 4: Cumulative instantaneous regret with parametric recovery functions.

## 8  Conclusion

In this work, we used Gaussian processes to model the recovering bandits problem and incorporated this into UCB and Thompson sampling algorithms. These algorithms use properties of GPs to look ahead and find good sequences of arms. They achieve $d$-step lookahead Bayesian regret of $O^*(\sqrt{KdT})$ for linear and squared exponential kernels, and perform well experimentally. We also improved computational efficiency of the algorithm using optimistic planning. Future work includes considering the frequentist setting, analyzing online methods for choosing $z_{\max}$ and the kernel, and investigating the use of GPs in other non-stationary bandit settings.

### Acknowledgments

CPB would like to thank the EPSRC funded EP/L015692/1 STOR-i centre for doctoral training and Sparx. We would also like to thank the reviewers for their helpful comments.

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
