[Supplementary Material]

# Supplementary Material for Recovering Bandits

## A   Preliminaries

Define the filtration $\{\mathcal{F}_t\}_{t=0}^{\infty}$ as $\mathcal{F}_0 = \emptyset$ and

$$\mathcal{F}_t = \sigma(J_1, \ldots, J_t, Y_1, \ldots, Y_t, \mathbf{Z}_1, \ldots, \mathbf{Z}_t) \tag{7}$$

where $\mathbf{Z}_t = [Z_{1,t}, \ldots, Z_{K,t}]$. It is important to note that $\mu_t(j), \sigma_t(j), J_t$ and $\mathbf{Z}_t$ are $\mathcal{F}_{t-1}$ measurable.

Recall that in both $d$RGP-UCB and $d$RGP-TS, we select a sequence of arms to play at time $t$ by building a $d$-step lookahead tree with root $\mathbf{Z}_t$ and selecting the leaf node $i$ with highest upper confidence bound on $M_i$, the cumulative reward from playing all arms in that sequence,

$$M_i(\mathbf{Z}_t) = \sum_{\ell=0}^{d-1} f_{J_{t+\ell}}(Z_{J_{t+\ell}, t+\ell})$$

where $\{J_{t+\ell}\}_{\ell=0}^{d-1}$ are the sequence of arms played on the path to leaf $i$ and $\{Z_{J_{t+\ell}, t+\ell}\}_{\ell=0}^{d-1}$ the corresponding $z$ values. Denote the posterior mean and variance of $M_i(\mathbf{Z}_t)$ at time $t$ as $\eta_t(i)$ and $\varsigma_t(i)$, then, conditional on the history $\mathcal{F}_{t-1}$, $M_i(\mathbf{Z}_t) \sim \mathcal{N}(\eta_t(i), \varsigma_t^2(i))$. When each arm can be played multiple times, there are interaction terms in the variance of the $M_i(\mathbf{Z}_t)$'s and thus we suffer some additional cost for not updating after every play. For each leaf node $i$, we can calculate

$$\varsigma_t^2(i) = \sum_{\ell=0}^{d-1} \sigma_t^2(J_{t+\ell}) + \sum_{\ell \neq q; \ell, q=0}^{d-1} \mathrm{cov}_t(f_{J_{t+\ell}}(Z_{J_{t+\ell}, t+\ell}), f_{J_{t+q}}(Z_{J_{t+q}, t+q}))$$

where $\mathrm{cov}_t(f_{J_{t+\ell}}(Z_{J_{t+\ell}, t+\ell}), f_{J_{t+q}}(Z_{J_{t+q}, t+q}))$ is $0$ if $J_{t+\ell} \neq J_{t+q}$ and $k_{J_{t+\ell}}(Z_{J_{t+\ell}, t+\ell}, Z_{J_{t+q} t+q}; N_{J_{t+\ell}}(t-1))$ for $J_{t+\ell} = J_{t+q}$. Note that throughout, we assume that the variances and covariances are calculated at the $Z_{j,t}$'s where the arms are played, ie. $\sigma_t^2(J_{t+\ell}) = \sigma_{J_{t+\ell}}^2(Z_{J_{t+\ell}, t+\ell}; N_{J_{t+\ell}}(t-1))$.

Before providing the proofs of the regret bounds, we need the following lemmas,

**Lemma 9**

$$\sum_{t=1}^{T} \sum_{j=1}^{K} \sigma_t^2(J_t) \mathbb{I}\{J_t = j\} \leq C_1 K \gamma_T.$$

*where $C_1 = 1/\log(1 + \sigma^{-2})$.*

*Proof:* Using the results of Lemma 5.4 of [28] and the fact that the maximal information gain is increasing in the number of data points, it follows that

$$\sum_{t=1}^{T} \sum_{j=1}^{K} \sigma_t^2(J_t) \mathbb{I}\{J_t = j\} = \sum_{j=1}^{K} \sum_{n=1}^{N_j(T)} \sigma_j^2(Z_j^{(n)}; n-1)$$

$$\leq T \sum_{j=1}^{K} C_1 I(\mathbf{y}_{j, N_j(T)}; \mathbf{f}_{j, N_j(T)}) \leq C_1 \sum_{j=1}^{K} \gamma_{N_j(T)} \leq C_1 K \gamma_T.$$

$\square$

The following lemmas bound the amount of information we loose by only updating the posterior every $d$ steps in the case where we can play each arm multiple times in a $d$-step lookahead. The result proves Lemma 4 in the main text.

**Lemma 10** *For any $z \in \mathcal{Z}$ arm $j$ and $n \in \mathbb{N}, n \geq 1$, let $Z^{(n)}$ be the $z$ value when arm $j$ is played for the $n$th time. Then,*

$$\sigma_j^2(z; n-1) - \sigma_j^2(z; n) = \frac{k_j^2(Z_j^{(n)}, z; n-1)}{\sigma_j^2(Z_j^{(n)}; n-1) + \sigma^2} \leq \frac{\sigma_j^2(Z_j^{(n)}; n-1)}{\sigma^2}$$

*Proof:* For convenience, we drop the $j$ notation and let $\mathbf{k}_n(z) = [k(Z^{(1)}, z), \ldots, k(Z^{(n)}, z)]^T$ and $\mathbf{K}_n = [k(Z^{(i)}, Z^{(j)})]_{i,j=1}^n$. Then,

$$
\begin{aligned}
\sigma^2&(z; n-1) - \sigma^2(z; n) \\
&= k(z, z) - \mathbf{k}_{n-1}(z)^T (\mathbf{K}_{n-1} + \sigma^2 \mathbf{I})^{-1} \mathbf{k}_{n-1}(z) - k(z, z) + \mathbf{k}_n(z)^T (\mathbf{K}_n + \sigma^2 \mathbf{I})^{-1} \mathbf{k}_n(z) \\
&= \mathbf{k}_n(z)^T (\mathbf{K}_n + \sigma^2 \mathbf{I})^{-1} \mathbf{k}_n(z) - \mathbf{k}_{n-1}(z)^T (\mathbf{K}_{n-1} + \sigma^2 \mathbf{I})^{-1} \mathbf{k}_{n-1}(z)
\end{aligned} \tag{8}
$$

We write,

$$
\mathbf{k}_n(z) = \begin{bmatrix} \mathbf{k}_{n-1}(z) \\ k(Z^{(n)}, z) \end{bmatrix} \qquad \mathbf{K}_n + \sigma^2 \mathbf{I} = \begin{pmatrix} \mathbf{K}_{n-1} + \sigma^2 \mathbf{I} & \mathbf{k}_{n-1}(z) \\ \mathbf{k}_{n-1}(z)^T & k(Z^{(n)}, Z^{(n)}) + \sigma^2 \end{pmatrix} = \begin{pmatrix} \mathbf{A} & \mathbf{B} \\ \mathbf{B}^T & C \end{pmatrix}.
$$

Then, by the block matrix inversion formula,

$$
(\mathbf{K}_n + \sigma^2 \mathbf{I})^{-1} = \begin{pmatrix} \mathbf{A}^{-1} + \mathbf{A}^{-1}\mathbf{B}(C - \mathbf{B}^T\mathbf{A}^{-1}\mathbf{B})^{-1}\mathbf{B}^T\mathbf{A}^{-1} & -\mathbf{A}^{-1}\mathbf{B}(C - \mathbf{B}^T\mathbf{A}^{-1}\mathbf{B})^{-1} \\ -(C - \mathbf{B}^T\mathbf{A}^{-1}\mathbf{B})^{-1}\mathbf{B}^T\mathbf{A}^{-1} & (C - \mathbf{B}^T\mathbf{A}^{-1}\mathbf{B})^{-1} \end{pmatrix}
$$

Hence,

$$
\begin{aligned}
\mathbf{k}_n(z)^T &(\mathbf{K}_n + \sigma^2 \mathbf{I})^{-1} \mathbf{k}_n(z) = [\mathbf{k}_{n-1}(z)^T, k(Z^{(n)}, z)](\mathbf{K}_n + \sigma^2 \mathbf{I})^{-1} \begin{bmatrix} \mathbf{k}_{n-1}(z) \\ k(Z^{(n)}, z) \end{bmatrix} \\
&= \mathbf{k}_{n-1}(z)^T (\mathbf{A}^{-1} + \mathbf{A}^{-1}\mathbf{B}(C - \mathbf{B}^T\mathbf{A}^{-1}\mathbf{B})^{-1}\mathbf{B}^T\mathbf{A}^{-1}) \mathbf{k}_{n-1}(z) \\
&\quad - k(Z^{(n)}, z)(C - \mathbf{B}^T\mathbf{A}^{-1}\mathbf{B})^{-1}\mathbf{B}^T\mathbf{A}^{-1}\mathbf{k}_{n-1}(z) \\
&\quad - \mathbf{k}_{n-1}(z)^T \mathbf{A}^{-1}\mathbf{B}(C - \mathbf{B}^T\mathbf{A}^{-1}\mathbf{B})^{-1} k(Z^{(n)}, z) \\
&\quad + k(Z^{(n)}, z)(C - \mathbf{B}^T\mathbf{A}^{-1}\mathbf{B})^{-1} k(Z^{(n)}, z) \\
&= \mathbf{k}_{n-1}(z)^T \mathbf{A}^{-1} \mathbf{k}_{n-1}(z) \\
&\quad + \mathbf{k}_{n-1}(z)^T (\mathbf{A}^{-1}\mathbf{B}(C - \mathbf{B}^T\mathbf{A}^{-1}\mathbf{B})^{-1}(\mathbf{B}^T\mathbf{A}^{-1}\mathbf{k}_{n-1}(z) - k(Z^{(n)}, z)) \\
&\quad + (k(Z^{(n)}, z) - \mathbf{k}_{n-1}(z)^T \mathbf{A}^{-1}\mathbf{B})(C - \mathbf{B}^T\mathbf{A}^{-1}\mathbf{B})^{-1} k(Z^{(n)}, z) \\
&= \mathbf{k}_{n-1}(z)^T \mathbf{A}^{-1} \mathbf{k}_{n-1}(z) \\
&\quad + (k(Z^{(n)}, z) - \mathbf{k}_{n-1}(z)^T \mathbf{A}^{-1}\mathbf{B})(C - \mathbf{B}^T\mathbf{A}^{-1}\mathbf{B})^{-1}(k(Z^{(n)}, z) - (\mathbf{k}_{n-1}(z)^T \mathbf{A}^{-1}\mathbf{B})^T)
\end{aligned}
$$

Then, substituting back $\mathbf{A} = \mathbf{K}_{n-1} + \sigma^2 \mathbf{I}, \mathbf{B} = \mathbf{k}_{n-1}(z), C = k(Z^{(n)}, z_{(n)}) + \sigma^2$ gives,

$$
\begin{aligned}
\mathbf{k}_n(z)^T (\mathbf{K}_n + \sigma^2 \mathbf{I})^{-1} \mathbf{k}_n(z) =\ &\mathbf{k}_{n-1}(z)^T (\mathbf{K}_{n-1} + \sigma^2 \mathbf{I})^{-1} \mathbf{k}_{n-1}(z) \\
&+ (k(Z^{(n)}, z) - \mathbf{k}_{n-1}(z)^T (\mathbf{K}_{n-1} + \sigma^2 \mathbf{I})^{-1} \mathbf{k}_{n-1}(z)) \\
&\quad (k(Z^{(n)}, Z^{(n)}) - \mathbf{k}_{n-1}(z_n)^T (\mathbf{K}_{n-1} + \sigma^2 \mathbf{I})^{-1} \mathbf{k}_{n-1}(z) + \sigma^2)^{-1} \\
&\quad (k(Z^{(n)}, z) - (\mathbf{k}_{n-1}(z)^T (\mathbf{K}_{n-1} + \sigma^2 \mathbf{I})^{-1} \mathbf{k}_{n-1}(z))^T) \\
=\ &\mathbf{k}_{n-1}(z)^T (\mathbf{K}_{n-1} + \sigma^2 \mathbf{I})^{-1} \mathbf{k}_{n-1}(z) + \frac{k^2(Z^{(n)}, z; n-1)}{\sigma^2(Z^{(n)}; n-1) + \sigma^2}
\end{aligned}
$$

Hence, substituting into (8) gives,

$$
\sigma^2(z; n-1) - \sigma^2(z; n) = \frac{k^2(Z^{(n)}, z; n-1)}{\sigma^2(Z^{(n)}; n-1) + \sigma^2}.
$$

Then, since the covariance matrix is positive semi-definite, for any $z, z'$ and $m \in \mathbb{N}$, $k(z, z'; m) \leq \sqrt{\sigma^2(z; m)\sigma^2(z'; m)}$ and so

$$
\sigma^2(z; n-1) - \sigma^2(z; n) \leq \frac{\sigma^2(Z^{(n)}; n-1)\sigma^2(z; n-1)}{\sigma^2(Z^{(n)}; n-1) + \sigma^2} \leq \frac{\sigma^2(Z^{(n)}; n-1)}{\sigma^2}
$$

since for any $z \in \mathcal{Z}$ and $m \in \mathbb{N}$, $0 \leq \sigma^2(z; m) \leq 1$. This concludes the proof. $\qquad \square$

We then use this result in the following lemma,

**Lemma 11** *For any leaf node $i$ of the $d$-step look ahead tree constructed at time $t$,*

$$\varsigma_t^2(i) \leq 3 \sum_{j=1}^{K} \left( \sum_{m=N_j(t)+1}^{N_j(t+d)} \frac{N_j(t+d) - m + 1}{\sigma^2} \sigma_j^2(z^{(m)}; m-1) \right) = \zeta_t^2$$

*and $\zeta_t$ is $\mathcal{F}_{t-1}$ measurable.*

*Proof:* First note that since the posterior covariance matrix of $f_j$ is positive semi-definite, for any $z_1, z_2$ and number of samples, $n-1$, $k_j(z_1, z_2; n-1) \leq 1/2(\sigma_j^2(z_1; n-1) + \sigma_j^2(z_2; n-1))$. Hence,

$$\varsigma_t(i) \leq 3 \sum_{\ell=0}^{d-1} \sigma_t^2(J_{t+\ell}).$$

Now consider arm $j$ and assume it appears $s \leq d$ times in the $d$-step look ahead policy selected at time $t$. Then, the contribution of arm $j$ (which for ease of notation we assume has been played $n-1$ times previously) to $\varsigma_t^2(i)$ is given below where we use the notation $\sigma_j^2(z^{(i)}; n-1)$ to denote the posterior variance at the $i$th $z$ of arm $j$ given $n-1$ observations of arm $j$.

$$\sum_{m=n}^{n+s-1} \sigma_j^2(Z_j^{(m)}; n-1) = \sigma_j^2(z^{(n)}; n-1) + \cdots + \sigma_j^2(z^{(n+s-1)}; n-1)$$

$$= \sigma_j^2(z^{(n)}; n-1) + \sigma_j^2(z^{(n+1)}; n) + \left( \sigma_j^2(z^{(n+1)}; n-1) - \sigma_j^2(z^{(n+1)}; n) \right) + \dots$$

$$+ \sigma_j^2(z^{(n+s-1)}; n+s-2) + \left( \sigma_j^2(z^{(n+s-1)}; n+s-3) - \sigma_j^2(z^{(n+s-1)}; n+s-2) \right)$$

$$+ \cdots + \left( \sigma_j^2(z^{(n+s-1)}; n-1) - \sigma_j^2(z^{(n+s-1)}; n) \right)$$

$$\leq \sigma_j^2(z^{(n)}; n-1) + \sigma_j^2(z^{(n+1)}; n) + \frac{\sigma_j^2(z^{(n)}; n-1)}{\sigma^2} + \cdots$$

$$+ \sigma_j^2(z^{(n+s-1)}; n+s-2) + \cdots + \frac{\sigma_j^2(z^{(n+1)}; n)}{\sigma^2} + \frac{\sigma_j^2(z^{(n)}; n-1)}{\sigma^2}$$

$$= \sum_{q=0}^{s-1} (1 + \frac{s-q-1}{\sigma^2}) \sigma_j^2(z^{(n+q)}; n+q-1)$$

$$\leq \sum_{q=0}^{s-1} \frac{s-q}{\sigma^2} \sigma_j^2(z^{(n+q)}; n+q-1)$$

which follows by recursively applying Lemma 4. Then, summing over all arms $j$ gives,

$$\varsigma_t^2(i) \leq 3 \sum_{j=1}^{K} \left( \sum_{m=N_j(t)+1}^{N_j(t+d)} \sigma_j^2(z^{(m)}; N_j(t)) \right)$$

$$\leq 3 \sum_{j=1}^{K} \left( \sum_{m=N_j(t)+1}^{N_j(t+d)} \frac{N_j(t+d) - m + 1}{\sigma^2} \sigma_j^2(z^{(m)}; m-1) \right)$$

Then, we note that $\zeta_t$ is $\mathcal{F}_{t-1}$ measurable since for a given leaf node $i$ of the tree constructed at time $t$, the sequence of arms played to get to node $i$ is known so $N_j(t+d)$ will be known and also the sequence of $Z_j^{(m)}$'s where arm $j$ is played will also be known. Since the posterior variance of arm $j$ after $m$ plays depends only on the number of plays and the covariates (not the observed rewards), $\sigma_j^2(z^{(m)}; m-1)$ is $\mathcal{F}_{t-1}$ measurable for $m = N_j(t) + 1, \ldots, N_j(t+d)$. $\square$

We also need the following result on the expectation of the maximum.

**Lemma 12** *Let $X_1, \ldots X_n$ be Gaussian random variables such that $\max_{1 \leq i \leq n} \mathbb{V}(X_i) \leq \zeta^2$. Then,*

$$\mathbb{E}\left[ \max_{1 \leq i \leq n} X_i \right] \leq \zeta \sqrt{2 \log(n)}.$$

*Proof:* See for example, Lemma 2.2 in [9]. $\square$

# B Theoretical Results for $d$RGP-UCB

We first prove the following lemma.

**Lemma 13** *For any leaf node $i$, initial node $z$ and constant $a > 0$,*

$$\int_a^\infty \mathbb{P}(M_i(z) - \eta_t(i) \geq x | \mathcal{F}_{t-1}) dx \leq \sqrt{2\pi} \varsigma_t(i) \exp\left\{-\frac{a^2}{2\varsigma_t^2(i)}\right\}.$$

*Proof:* The proof follows using the normality of the posterior of $M_i(z)$ (so at time $t$, $M_i(\mathbf{Z}_t) \sim \mathcal{N}(\eta_t(i), \varsigma_t(i)^2)$).

$$\begin{aligned}
\int_a^\infty \mathbb{P}(M_i(z) - \eta_t(i) \geq x | \mathcal{F}_{t-1}) dx &\leq \int_a^\infty \exp\left\{-\frac{x^2}{2\varsigma_t^2(i)}\right\} dx \\
&= \sqrt{2\pi} \varsigma_t(i) \int_a^\infty \frac{1}{\sqrt{2\pi}\varsigma_t(i)} \exp\left\{-\frac{x^2}{2\varsigma_t^2(i)}\right\} dx \\
&\leq \sqrt{2\pi} \varsigma_t(i) \exp\left\{-\frac{a^2}{2\varsigma_t^2(i)}\right\}.
\end{aligned}$$

Where we have used that if $X \sim \mathcal{N}(\mu, \sigma^2)$, $\mathbb{P}(X - \mu \geq b) \leq \exp\{-\frac{b^2}{2\sigma^2 2}\}$ for any $b > 0$, and the last inequality follows through integration of the pmf of a $\mathcal{N}(0, \varsigma_t(i))$ random variable. □

Then, define $M_{I_t^*}(\mathbf{Z}_t)$ to be the sum of the $f_j(z)$'s to leaf $I_t^*$ of the optimal $d$ step look ahead policy from time $t$ chosen using the unknown $f_j(z)$'s. Let $r_t$ be the per step regret at time $t$. We now bound the expected regret from time steps $t, t+1, \ldots, t+d-1$ where we have played arms according to the choice of $I_t$ by our algorithm. Let $r_s$ be the contribution to the regret at time $s$, that is $r_s = f_{J_t^*}(Z_{J_t^*, t}) - f_{J_t}(Z_{J_t, t})$. Then, let

$$\alpha_t = \sqrt{2\log((K|\mathcal{Z}|)^d (t+d-1)^2)}.$$

We will use the following lemma,

**Lemma 14** *Assume we start a $d$-step look ahead policy at time $t$, selecting leaf node $I_t$, then*

$$\sum_{s=t}^{t+d-1} \mathbb{E}[r_s | \mathcal{F}_{t-1}] \leq \frac{\sqrt{2d\pi}}{(t+d-1)^2} + \alpha_t \varsigma_t(I_t).$$

*Proof:* From (3), the upper confidence bound of node $i$ at time $t$ is given by,

$$\eta_t(i) + \alpha_t \varsigma_t(i),$$

and since we play node $I_t$, this has the highest upper confidence bound. Then, we use the following decomposition of the regret,

$$\begin{aligned}
\sum_{s=t}^{t+d-1} \mathbb{E}[r_s | \mathcal{F}_{t-1}] &= \mathbb{E}[M_{I_t^*}(\mathbf{Z}_t) - M_{I_t}(\mathbf{Z}_t) | \mathcal{F}_{t-1}] \\
&= \mathbb{E}[M_{I_t^*}(\mathbf{Z}_t) - (\eta_t(I_t^*) + \alpha_t \varsigma_t(I_t^*)) + (\eta_t(I_t^*) + \alpha_t \varsigma_t(I_t^*)) - M_{I_t}(\mathbf{Z}_t) | \mathcal{F}_{t-1}] \\
&\leq \mathbb{E}[M_{I_t^*}(\mathbf{Z}_t) - (\eta_t(I_t^*) + \alpha_t \varsigma_t(I_t^*)) + (\eta_t(I_t) + \alpha_t \varsigma_t(I_t)) - M_{I_t}(\mathbf{Z}_t) | \mathcal{F}_{t-1}] \\
&= \mathbb{E}[M_{I_t^*}(\mathbf{Z}_t) - \eta_t(I_t^*) - \alpha_t \varsigma_t(I_t^*) | \mathcal{F}_{t-1}] + \mathbb{E}[\eta_t(I_t) + \alpha_t \varsigma_t(I_t) - M_{I_t}(\mathbf{Z}_t) | \mathcal{F}_{t-1}]
\end{aligned}$$

For the first term, note that for any random variable $X$, $\mathbb{E}[X] \leq \mathbb{E}[X\mathbb{I}\{X > 0\}] = \int_0^\infty \mathbb{P}(X \geq x) dx$. Then, by Lemma 13 and using the fact that $\varsigma_t^2(i) \leq \sum_{\ell=0}^{d-1} k(z_\ell, z_\ell) \leq d$, it follows that,

$$\begin{aligned}
\mathbb{E}[M_{I_t^*}(\mathbf{Z}_t) - \eta_t(I_t^*) - \alpha_t \varsigma_t(I_t^*) | \mathcal{F}_{t-1}] &\leq \int_0^\infty \mathbb{P}(M_{I_t^*}(\mathbf{Z}_t) - \eta_t(I_t^*) - \alpha_t \varsigma_t(I_t^*) \geq x | \mathcal{F}_{t-1}) dx \\
&\leq \int_0^\infty \sum_{i=1}^{K^d} \sum_{z \in \mathcal{Z}^d} \mathbb{P}(M_i(z) - \eta_t(i) - \alpha_t \varsigma_t(i) \geq x | \mathcal{F}_{t-1}) dx
\end{aligned}$$

$$= \sum_{i=1}^{K^d} \sum_{z \in \mathcal{Z}^d} \int_{\alpha_t \varsigma_t(i)}^{\infty} \mathbb{P}(M_i(z) - \eta_t(i) \geq x | \mathcal{F}_{t-1}) dx$$

$$= \sum_{i=1}^{K^d} \sum_{z \in \mathcal{Z}^d} \sqrt{2\pi} \varsigma_t(i) \exp\left\{ -\frac{(\alpha_t \varsigma_t(i))^2}{2\varsigma_t^2(i)} \right\}$$

$$\leq \sum_{i=1}^{K^d} \sum_{z \in \mathcal{Z}^d} \sqrt{2d\pi} \frac{1}{(t+d-1)^2 (K|\mathcal{Z}|)^d}$$

$$= \frac{\sqrt{2d\pi}}{(t+d-1)^2},$$

where the last inequality follows from the definition of $\alpha_t$.

For the second term, recall that $\eta_t(i) = \mathbb{E}[M_i(\mathbf{Z}_t)|\mathcal{F}_{t-1}]$ and $I_t$ is $\mathcal{F}_{t-1}$ measurable. Hence,

$$\mathbb{E}[\eta_t(I_t) + \alpha_t \varsigma_t(I_t) - M_{I_t}(\mathbf{Z}_t)|\mathcal{F}_{t-1}] = \eta_t(I_t) + \alpha_t \varsigma_t(I_t) - \eta_t(I_t) = \alpha_t \varsigma_t(I_t).$$

Combining both terms gives the result.

$\square$

We now prove the regret bounds for $d$RGP-UCB in the repeating and non-repeating cases.

### B.1 Non-Repeating

**Theorem 2** *The d-step single play lookahead regret of dRGP-UCB satisfies,*

$$\mathbb{E}[\mathfrak{R}_T^{(d,s)}] \leq O(\sqrt{KT\gamma_T \log(TK|\mathcal{Z}|)}).$$

*Proof:* For ease of notation define $\mathfrak{R}_T$ as the $d$-step lookahead regret with single plays that we are interested in (i.e. $\mathfrak{R}_T = \mathfrak{R}_T^{(d,s)}$) and note that,

$$\mathbb{E}[\mathfrak{R}_T] \leq \sum_{h=0}^{\lfloor T/d \rfloor} \mathbb{E}\left[ \sum_{s=hd+1}^{(h+1)d} \mathbb{E}[r_s|\mathcal{F}_{hd}] \right].$$

Then, using Lemma 14, and the fact that since we cannot repeat plays, $\sigma_t(J_{t+\ell}) = \sigma_{t+\ell}(J_{t+\ell})$ for any $\ell = 0, \ldots, d-1$,

$$\mathbb{E}[\mathfrak{R}_T] \leq \sum_{h=0}^{\lfloor T/d \rfloor} \mathbb{E}\left[ \sum_{s=hd+1}^{(h+1)d} \mathbb{E}[r_s|\mathcal{F}_{hd}] \right]$$

$$\leq \sum_{h=0}^{\lfloor T/d \rfloor} \mathbb{E}\left[ \frac{\sqrt{2d\pi}}{(h+1)^2 d^2} + \alpha_{hd+1} \sqrt{\varsigma_{hd+1}^2(I_{hd+1})} \right]$$

$$\leq \frac{\sqrt{2\pi}}{d} \sum_{h=1}^{\lfloor T/d \rfloor+1} \frac{1}{h^2} + \sum_{h=0}^{\lfloor T/d \rfloor} \sqrt{2\log((K|\mathcal{Z}|)^d (h+1)^2 d^2)} \mathbb{E}\left[ \sqrt{\sum_{\ell=0}^{d-1} \sigma_{hd+1}(J_{hd+1+\ell})} \right]$$

$$\leq \frac{\pi^{5/2}}{\sqrt{2}3d} + \sqrt{4\log((K|\mathcal{Z}|)^d (T+d))} \sqrt{\lfloor T/d \rfloor + 1} \mathbb{E}\left[ \sqrt{\sum_{t=1}^{T} \sigma_t^2(J_t)} \right]$$

$$\leq \frac{\pi^{5/2}}{\sqrt{2}3d} + \sqrt{4\log((K|\mathcal{Z}|)^d (T+d))} \sqrt{\lfloor T/d \rfloor + 1} \mathbb{E}\left[ \sqrt{\sum_{j=1}^{K} \sum_{t=1}^{T} \sigma_t^2(j) \mathbb{I}\{J_t = j\}} \right]$$

$$\leq \frac{\pi^{5/2}}{\sqrt{2}3d} + \sqrt{4\log((K|\mathcal{Z}|)^d (T+d))} \sqrt{\lfloor T/d \rfloor + 1} \sqrt{C_1 K \gamma_T}$$

where $C_1 = 1/\log(1 + \sigma^{-2})$ and the last line follows by Lemma 9. This gives the result.

$\square$

## B.2 Repeating

**Theorem 5** *The $d$-step multiple play lookahead regret of dRGP-UCB satisfies,*

$$\mathbb{E}[\mathfrak{R}_T^{(d,m)}] \leq O\left(\sqrt{KT\gamma_T \log((K|\mathcal{Z}|)^d T)}\right).$$

*Proof:* For ease of notation define $\mathfrak{R}_T$ as the $d$-step lookahead regret with multiple plays that we are interested in (i.e. $\mathfrak{R}_T = \mathfrak{R}_T^{(d,m)}$) and note that,

$$\mathbb{E}[\mathfrak{R}_T] = \sum_{h=0}^{\lfloor T/d \rfloor} \mathbb{E}\left[\sum_{s=hd+1}^{(h+1)d} \mathbb{E}[r_s|\mathcal{F}_{hd}]\right].$$

Then, note that from Lemma 11, it follows that

$$\varsigma_t^2(i) \leq 3 \sum_{j=1}^{K} \left(\sum_{m=N_j(t)+1}^{N_j(t+d)} \frac{N_j(t+d)-m+1}{\sigma^2}\sigma_j^2(z^{(m)};m-1)\right) \leq \frac{3d}{\sigma^2}\sum_{j=1}^{K}\sum_{m=N_j(t)+1}^{N_j(t+d)}\sigma_j^2(z^{(m)};m-1).$$

Hence, by lemma 14 and summing over all time points where we start a $d$-step look ahead policy, it follows that,

$$\mathbb{E}[\mathfrak{R}_T] = \sum_{h=0}^{\lceil T/d \rceil-1} \mathbb{E}\left[\sum_{s=hd+1}^{(h+1)d} \mathbb{E}[r_s|\mathcal{F}_{hd}]\right]$$

$$\leq \sum_{h=0}^{\lfloor T/d \rfloor} \mathbb{E}\left[\frac{\sqrt{2d\pi}}{(h+1)^2 d^2} + \alpha_{hd+1}\sqrt{\varsigma_{hd+1}^2(I_{hd+1})}\right]$$

$$\leq \frac{\sqrt{2\pi}}{d}\sum_{h=1}^{\lfloor T/d \rfloor+1}\frac{1}{h^2} + \sum_{h=0}^{\lfloor T/d \rfloor}\sqrt{2\log((K|\mathcal{Z}|)^d(h+1)^2 d^2)}\mathbb{E}\left[\sqrt{\frac{3d}{\sigma^2}\sum_{j=1}^{K}\sum_{m=N_j(dh)+1}^{N_j(d(h+1))}\sigma_j^2(z^{(m)};m-1)}\right]$$

$$\leq \frac{\pi^{5/2}}{\sqrt{23}d} + \sqrt{\frac{12d}{\sigma^2}\log((K|\mathcal{Z}|)^d(T+d))}\sqrt{\lfloor T/d \rfloor+1}\mathbb{E}\left[\sqrt{\sum_{h=0}^{\lfloor T/d \rfloor}\sum_{j=1}^{K}\sum_{m=N_j(dh)+1}^{N_j(d(h+1))}\sigma_j^2(z^{(m)};m-1)}\right]$$

Then, from Lemma 9 and the fact that $\gamma_n$ is increasing in $n$,

$$\sqrt{\sum_{h=0}^{\lfloor T/d \rfloor}\sum_{j=1}^{K}\sum_{m=N_j(dh)+1}^{N_j(d(h+1))}\sigma_j^2(z^{(m)};m-1)} = \sqrt{\sum_{j=1}^{K}\sum_{m=1}^{N_j(T)}\sigma_j^2(z^{(m)};m-1)}$$

$$\leq \sqrt{\sum_{j=1}^{K}C_1\gamma_{N_j(T)}} \leq \sqrt{C_1 K\gamma_T}$$

for $C_1 = (1+\log(\sigma^{-2}))^{-1}$. Hence,

$$\mathbb{E}[\mathfrak{R}_T] \leq \frac{\pi^{5/2}}{\sqrt{23}d} + \sqrt{\frac{12d}{\sigma^2}\log((K|\mathcal{Z}|)^d(T+d))}\sqrt{T/d+1}\sqrt{C_1 K\gamma_T}$$

and so the result follows. $\qquad\square$

# C  Theoretical Results for $d$RGP-TS

The regret bounds for the Thompson sampling approach($d$RGP-TS) follow in a similar manner to those for $d$RGP-UCB using the techniques of [25]. Specifically, using [25], we get the following result which is equivalent to Lemma 14, and which can then be used to get the regret bound much in the same way as Theorem 2 and Theorem 5.

**Lemma 15** *Assume we start a $d$-step look ahead policy at time $t$, selecting leaf node $I_t$, then*

$$\sum_{s=t}^{t+d-1} \mathbb{E}[r_s|\mathcal{F}_{t-1}] \leq \frac{\sqrt{2d\pi}}{(t+d-1)^2} + \alpha_t\varsigma_t(I_t).$$

*Proof:* As in [25] we relate the Bayesian regret of Thompson sampling to the upper confidence bounds used in our upper confidence bound approach. Specifically, by Proposition 1 in [25],

$$\sum_{s=t}^{t+d-1} \mathbb{E}[r_s|\mathcal{F}_{t-1}] = \mathbb{E}[M_{I_t^*}(\mathbf{Z}_t) - M_{I_t}(\mathbf{Z}_t)|\mathcal{F}_{t-1}]$$

$$= \mathbb{E}[M_{I_t^*}(\mathbf{Z}_t) - \eta_t(I_t^*) - \alpha_t\varsigma_t(I_t^*)|\mathcal{F}_{t-1}] + \mathbb{E}[\eta_t(I_t) + \alpha_t\varsigma_t(I_t) - M_{I_t}(\mathbf{Z}_t)|\mathcal{F}_{t-1}]$$

The same argument as Lemma 14 then gives the result.

$\square$

## C.1 Non-Repeating

**Theorem 3** *The $d$-step single play lookahead regret of dRGP-TS satisfies,*

$$\mathbb{E}[\mathfrak{R}_T^{(d,s)}] \leq O(\sqrt{KT\gamma_T \log(TK|\mathcal{Z}|)}).$$

*Proof:* Given Lemma 15, the proof follows in the same manner as the proof of Theorem 2. $\square$

## C.2 Repeating

**Theorem 6** *The $d$-step multiple play lookahead regret of dRGP-TS satisfies,*

$$\mathbb{E}[\mathfrak{R}_T^{(d,m)}] \leq O\left(\sqrt{KT\gamma_T \log((K|\mathcal{Z}|)^d T)}\right).$$

*Proof:* Again, the proof follows by the same argument as Theorem 5 using Lemma 15. $\square$

# D Optimality of the Lookahead Oracle

For any policy $\pi$, let $V_T(\pi)$ denote the expected cumulative reward from playing policy $\pi$ up to horizon $T$. We say a policy $\pi$ is periodic with period $p \in \mathbb{N}$ from some initial $\mathbf{z}_1$ if there is some $t_0 > 0$ such that for all $t > t_0$, $\mathbf{z}_t^\pi = \mathbf{z}_{t+p}^\pi$ and $j_t^\pi = j_{t+p^*}^\pi$, where $j_t^\pi$ is the action taken at time $t$ by policy $\pi$ and $\mathbf{z}_t^\pi$ is the vector of $z$ values obtained at time $t$ from playing according to policy $\pi$ for $t-1$ steps. For a periodic policy $\pi$ and initial $\mathbf{z}_1$, we will assume that $p \geq t_0$.

**Lemma 16** *If $\pi^*$ is an optimal stationary deterministic policy then if $T > |\mathcal{Z}|^K$, then $\pi^*$ must be periodic with some period $p^* \leq |\mathcal{Z}|^K$.*

*Proof:* The proof follows by noting that $\pi^*$ must be a deterministic mapping from $\mathbf{z}$ to actions since a stationary policy does not depend on the time step. In particular, $\pi^* : \mathcal{Z}^K \to \{1, \ldots, K\}$ with $\pi^*(\mathbf{z}) = j$ for some $1 \leq j \leq K$, and each $\mathbf{z}$ corresponds to only one action. We now argue for a contradiction. Assume that $\pi^*$ is not periodic. Then since $T > |\mathcal{Z}|^K$, there must exist some $\mathbf{z}$ which is arises twice, so there exists some $t$ and $0 < p^* \leq |\mathcal{Z}|^K$ such that $\mathbf{z}_t = \mathbf{z}_{t+p^*} = \mathbf{z}$. Since $\pi^*$ is a deterministic mapping, the same action must be taken in both cases, which will lead to the same next value of $\mathbf{z}' = \mathbf{z}_{t+1} = \mathbf{z}_{t+p^*+1}$, since the evolution of the $\mathbf{z}$ is deterministic conditional on actions. Repeatedly applying same argument, we see that $\pi^*$ will take the same sequence of $p^*$ actions from $\mathbf{z}$ in both cases before returning to $\mathbf{z}$ (if the horizon is long enough). Hence $\pi^*$ must be periodic, contradicting the assumption. $\square$

**Proposition 1** *Let $p^*$ be the period of the optimal SD policy $\pi^*$. For any $l = 1, \ldots, \lfloor \frac{T-z_{\max}}{p^*} \rfloor$, the optimal $(z_{\max} + lp^*)$-lookahead policy, $\pi_l^*$, satisfies, $V_T(\pi_l^*) \geq \left(1 - \frac{(l+1)p^* + z_{\max}}{T+p^*}\right) \frac{lp^*}{lp^* + z_{\max}} V_T(\pi^*).$*

*Proof:* Define a vector $\mathbf{z} = (z_1, \ldots, z_K)$ as feasible for the recovering bandits problem starting from $\mathbf{z}_0$ with $K$ arms and a fixed value of $z_{\max}$, if it is possible to play a sequence of arms up to any time $t \geq 1$ such that $\mathbf{z}_t = \mathbf{z}$. We begin by observing that it is possible to get from any feasible $\mathbf{z} = (z_1, \ldots, z_K)$ to any other feasible $\mathbf{z}' = (z'_1, \ldots, z'_K)$ in at most $z_{\max}$ steps. For this, we need the following properties of $\mathbf{z}$ that are consequences of the update procedure in equation (1). Equation (1) guarantees that there must be exactly one element of $\mathbf{z}$ equal to 0, and if $z_i, z_j \neq z_{\max}$, then $z_i \neq z_j$ for $i \neq j$. For the target vector $\mathbf{z}'$, let $n$ be the number of elements with value $z_{\max}$. The remaining $K - n$ entries must all be unique and one must be 0, denote the index of this $i_0$. In the following $z_{\max}$ steps, we play each arm corresponding to $z_i \neq z_{\max}$ at step $z_{\max} - z_i$ and play $i_0$ in the intervening steps, and at step $z_{\max}$. It is clear to see that this procedure will go from $\mathbf{z}$ to $\mathbf{z}'$ in $z_{\max}$ steps.

Let $v^*$ be the reward achieved in $p^*$ steps of the optimal policy $\pi^*$. By the above argument, from any initial state of the lookahead $\mathbf{Z}_t$, it is possible to get to any other (feasible) $\mathbf{z}$ in at most $z_{\max}$ steps. In particular, it is possible to get to one of the elements $\mathbf{z}^{(1)}, \ldots, \mathbf{z}^{(p^*)}$ of the optimal periodic policy in $z_{\max}$ steps. Hence, the policy that chooses the quickest route to the optimal periodic policy and then plays that policy for $lp^*$ steps is a valid $(z_{\max} + lp^*)$-lookahead policy. This policy will achieve reward of at least $lv^*$ over this period. Consequently, the optimal $(z_{\max} + lp^*)$-lookahead policy, $\pi^*_l$ will achieve reward of at least $lv^*$ every $(z_{\max} + lp^*)$ steps. We select a lookahead policy every $(z_{\max} + lp^*)$ steps, therefore the total reward of $\pi^*_l$ must be at least $\lfloor \frac{T}{lp^* + z_{\max}} \rfloor lv^*$. The total reward of $\pi^*$ is less than $\lceil \frac{T}{p^*} \rceil v^*$. Therefore,

$$\frac{V_T(\pi^*_l)}{V_T(\pi^*)} \geq \frac{\lfloor \frac{T}{lp^* + z_{\max}} \rfloor lv^*}{\lceil \frac{T}{p^*} \rceil v^*} \geq \frac{\frac{T}{lp^* + z_{\max}} - 1}{\frac{T}{p^*} + 1} l = \left(1 - \frac{(l+1)p^* + z_{\max}}{T + p^*}\right) \frac{lp^*}{lp^* + z_{\max}}.$$

This gives the result. $\qquad\square$

## E  Theoretical Guarantees on Optimistic Planning Procedure

**Proposition 8** *For the optimistic planning procedure with budget $N$, if the procedure stops at step $n < N$ because a node $i_n$ of depth $d$ is selected, then $v^* - v(i_n) = 0$. Otherwise, if there exist $\lambda \in (\frac{1}{K}, 1]$ and $1 \leq d_0 \leq d$ such that $\forall l \geq d_0$, $p_l((d - l)\Delta) \leq \lambda^l$, then for $N > n_0 = \frac{K^{d_0+1} - 1}{K - 1}$,*

$$v^* - v(i_N) \leq \left(d - \frac{\log(N - n_0)}{\log(\lambda K)} - \frac{\log(\lambda K - 1)}{\log(\lambda K)} + 1\right)\Delta. \tag{6}$$

*Proof:* Since our $\tilde{f}_j(z)$'s are samples from a Gaussian posterior, they can be negative. Hence it will be convenient to work with a transformation that guarantees positivity. To this end, let $\delta = -\min_{j,z} \tilde{f}_j(z)$ if $\min_{j,z} \tilde{f}_j(z) < 0$ and $\delta = 0$ if $\min_{j,z} \tilde{f}_j(z) \geq 0$ and for any arm $j$ and covariate $z$, define,

$$\tilde{f}'_j(z) = \tilde{f}_j(z) + \delta \geq 0.$$

Then we define the corresponding $v$, $b$ and $u$ values of any node $i \in \mathcal{S}_n$ at step $n$ and $\Psi$ functions as,

$$v'(i) = v(i) + d\delta \qquad b'_n(i) = b_n(i) + d\delta \qquad u'(i) = u(i) + l(i)\delta$$
$$\Psi'(\mathbf{z}(i), d - l(i)) = \Psi(\mathbf{z}(i), d - l(i)) + (d - l(i))\delta \qquad \Psi'^*(l) = \Psi^*(l) + l\delta,$$

where $l(i)$ is the depth of node $i$. Note that node $i^*$ maximizing $v(i)$ will also maximize $v'(i)$ and that if at step $n$ we select a node maximizing $b_n(i)$ this will also be the node maximizing $b'_n(i)$ and so $v(i_1) \geq v(i_2) \iff v'(i_1) \geq v'(i_2)$ and $b(i_1) \geq b(i_2) \iff b'(i_1) \geq b'(i_2)$ for all nodes $i_1, i_2$. Furthermore, it holds that $v'(i) \geq u'(i)$ and that $b'(i)$ is an upper bound on $v'(i)$ for all nodes $i$ and in particular $b'(i) = u'(i) + \Psi'(\mathbf{z}(i), d - l(i))$.

We begin with the case where the algorithm is stopped after some number $n$ of nodes have been expanded because the selected node is of depth $d$. Let $i^*_1, \ldots, i^*_d$ be the nodes on the path to the optimal node $i^*$ and let $j$ be the maximal depth of this path in $\mathcal{T}_n \cup \mathcal{S}_n$. If $i_n$ is the node at depth $d$ selected to be expanded at time $n$, then,

$$0 \leq v^* - v(i_n) = v'(i^*_j) - v'(i_n) \leq b'(i^*_j) - v'(i_n) \leq b'(i_n) - v'(i_n) = \Psi'(\mathbf{z}(i_n), d - d) = 0,$$

since we select node $i_n$ at time $n$ so it must have the largest $b_n(i)$ and $b'_n(i)$ value. This proves the first statement.

For the other case, define the set

$$\Gamma = \bigcup_{l=0}^{d} \{ \text{ node } i \text{ of depth } l \text{ such that } v^* - v(i) \leq \Psi'^*(d-l) \},$$

and note that if $v^* - v(i) \leq \Psi'^*(d-l)$ then also $v'^* - v'(i) \leq \Psi'^*(d-l)$. As in [13], we will show that all nodes expanded by our algorithm are in $\Gamma$. For this, let node $i$ of depth $l$ be chosen to be expanded at time $n$. This means it has the largest $b_n(i)$ (and $b'_n(i)$) value of all nodes in $\mathcal{S}_n$. We also now need to define the $b$ value of a node in $\mathcal{T}_n$ as $b_n(i) = \max_{j \in C(i)} b_n(j)$ where $C(i)$ is the set of all children of node $i$, and we define $b'_n(i)$ correspondingly. This definition together with the previous remark means that for any $j \in \mathcal{T}_n$, $b'_n(i) \geq b'_n(j)$. Then for some $1 \leq j \leq d$, $i_j^* \in \mathcal{T}_n$, so it follows that $b'_n(i_j^*) \leq b'_n(i_n)$. But, the best value of any continuation of a path to the optimal node is simply $v^*$ and so by definition of the $b$ values $b'_n(i_j^*) \geq v'(i_j^*) = v'^*$. Hence, since $v'(i) \geq u'(i)$ and $\Psi'(\mathbf{z}(i), d-l) \leq \Psi'^*(d-l)$,,

$$v'(i) \geq u'(i) = b'_n(i) - \Psi'(\mathbf{z}(i), d-l) \geq b'_n(i_j^*) - \Psi'(\mathbf{z}(i), d-l) \geq v'^* - \Psi'(\mathbf{z}(i), d-l) \geq v'^* - \Psi'^*(d-l),$$

it follows that $i \in \Gamma$. Then, we bound from below the maximal depth at which a node is chosen to be expanded. Let $n_0$ be the number of policies in $\Gamma$ up to depth $d_0$ and let $d_N$ be the maximal depth of any node expanded before the algorithm is stopped at time $N$. By the assumption in the proposition, the proportion of $(d-l)\Delta$-optimal nodes at depth $l$ is bounded by $\lambda^l$. Then, $\Psi'^*(d-l) = \Psi(d-l) + (d-l)\delta \leq (d-l) \max_{j,z} \tilde{f}_j(z) - (d-l) \min_{j,z} \tilde{f}_j(z) = (d-l)\Delta$ by definition of $\Psi$ and so $p_l(\Psi'^*(d-l)) \leq p_l((d-l)\Delta) \leq \lambda^l$. Hence,

$$N \leq n_0 + \sum_{l=d_0}^{d_N} \lambda^l K^l = n_0 + \sum_{l=d_0}^{d_N} A^l \leq n_0 + A^{d_0+1} \frac{A^{d_N - d_0} - 1}{A - 1}$$

for $A = \lambda K > 1$. Rearranging gives,

$$d_N \geq d_0 + \log_A \left( \frac{(N - n_0)(A-1)}{A^{d_0+1}} + 1 \right) \geq d_0 + \log_A \left( \frac{(N - n_0)(A-1)}{A^{d_0+1}} \right)$$

$$\geq \frac{\log(N - n_0)}{\log(K\lambda)} - 1 + \frac{\log(\lambda K - 1)}{\log(\lambda K)}$$

Let $i_N$ be the node the algorithm outputs at step $N$ when the computational resources have been exceeded and note that this is the node in $\mathcal{T}_N$ with largest depth (i.e. $l(i_N) = d_N$) that has the largest $b_N$ (or $b'_N$) value. Since $i_N \in \mathcal{T}_N$, there is some step $n \leq N$ when node $i_N$ was expanded. Then, let $j$ be the maximal depth of nodes on the path $i_1^*, \ldots, i_d^*$ in $\mathcal{S}_n$. It then follows that

$$v'^* - v'(i_N) \leq b'_n(i_j^*) - v'(i_N) \leq b'_n(i_N) - v(i_N) \leq \Psi'(\mathbf{z}(i_N), d - l(i_N)) \leq \Psi'^*(d - d_N).$$

Hence,

$$v^* - v(i_N) = v'^* - v'(i_N) \leq \Psi'^*(d - d_N) = \Psi^*(d - d_N) + (d - d_N)\delta$$

$$\leq (d - d_N)(\max_{j,z} \tilde{f}_j(z) - \min_{j,z} \tilde{f}_j(z)) \leq \left( d - \frac{\log(N - n_0)}{\log(K\lambda)} - \frac{\log(\lambda K - 1)}{\log(\lambda K)} + 1 \right) \Delta$$

which gives the result.

$\square$

# F  Regret Bounds for Non-Parametric Approach

We use an algorithm which has no information about the recovery structure as a baseline. For this, we model each (arm, $z$) pair as an arm. This reduces the problem to a standard multi-armed bandit problem with $K|\mathcal{Z}|$ arms, where only some arms are available each round.

Let $\mu_{j,z}$ denote the expected reward of arm $j$ when $z_j = z$. We can then create estimates $\bar{Y}_{j,z,t}$ of the reward of each arm from the $N_{j,z}(t)$ samples of arm $j$ with $Z_j = z$ we receive up to time $t$.

These estimates can be used to define an upper confidence bound style algorithm over the 'arms' $\{(j, z)\}_{j=1, z=0}^{K, Z_{\max}}$. We define confidence bound based on UCB1 [2] and [25]

$$U(j, z, t) = \bar{Y}_{z,j,t} + \sqrt{\frac{\sigma^2(2 + 6\log(T))}{N_{j,z}(t)}}.$$

where $\sigma$ is the standard error of the noise. After playing each $j, z$ combinations once, we proceed to play the arm with largest $U(j, Z_{j,t}, t)$ at time $t$. We now bound the Bayesian regret of this algorithm to horizon $T$.

**Theorem 17** *The instantaneous regret up to time $T$ of the UCB1 algorithm with $K|\mathcal{Z}|$ arms can be bounded by*

$$\mathbb{E}[\mathfrak{R}_T^{(1)}] \leq O(\sqrt{K|\mathcal{Z}|T\log(T)} + K|\mathcal{Z}|^2)$$

*Proof:* We first consider the initialization phase. For this, note that in order to play arm $j$ at $Z_j = z$, we need to wait $z$ rounds from when it was last played. This means that the total number of plays required to play each arm at each $z$ value can be bounded by $t_0 = K|\mathcal{Z}|(|\mathcal{Z}| + 1)$ (since in the worst case, for arm $j$, we need to wait, 1 round, then 2 rounds, up to $|\mathcal{Z}|$ rounds). We can bound the per-step regret from this initialization period using Lemma 12. For any $1 \leq t \leq t_0$,

$$\mathbb{E}[f_{J_t^*}(Z_{J_t^*,t}) - f_{J_t}(Z_{J_t,t})] \leq \mathbb{E}[\max_{1 \leq t \leq t_0}\{f_{J_t^*}(Z_{J_t^*,t}) - f_{J_t}(Z_{J_t,t})\}] \leq 2\sqrt{2\log(t_0)}$$

since the distribution of the difference of two zero mean Gaussian random variables with variances $\sigma_1^2, \sigma_2^2 \leq 1$ is also a Gaussian random variable with mean 0 and variance $\sigma_1^2 + \sigma_2^2 \leq 2$ here. Then, we can use a similar technique to [25] to bound the cumulative regret in the remaining $t_0 + 1 \leq t \leq T$ steps but using Lemma 12 again to bound the maximal difference in $f_j$'s.

$$\mathbb{E}[\mathfrak{R}_T] = \sum_{t=t_0}^{T} \mathbb{E}[f_{J_t^*}(Z_{J_t^*,t}) - f_{J_t}(Z_{J_t,t})\mathbb{I}\{\forall j, z; f_j(z) \in [L(j, z, t), U(j, z, t)]\}]$$

$$+ \sum_{t=t_0}^{T} \mathbb{E}[f_{J_t^*}(Z_{J_t^*,t}) - f_{J_t}(Z_{J_t,t})\mathbb{I}\{\exists j, z; f_j(z) \notin [L(j, z, t), U(j, z, t)]\}]$$

$$\leq \sum_{t=t_0}^{T} \mathbb{E}[U(J_t^*, Z_{J_t^*,t}, t) - L(J_t, Z_{J_t,t}, t)] + 2\sqrt{2\log(T)}T\mathbb{P}(\exists j, z; f_j(z) \notin [L(j, z, t), U(j, z, t)])$$

$$\leq \sum_{t=t_0}^{T} \mathbb{E}[U(J_t, Z_{J_t,t}, t) - L(J_t, Z_{J_t,t}, t)] + 2\sqrt{2\log(T)}T\sum_{j=1}^{K}\sum_{z \in \mathcal{Z}}\mathbb{P}(f_j(z) \notin [L(j, z, t), U(j, z, t)])$$

Since $\epsilon_t \sim \mathcal{N}(0, \sigma^2)$, by Lemma 1 in [25],

$$2\sqrt{2\log(T)}T\sum_{j=1}^{K}\sum_{z \in \mathcal{Z}}\mathbb{P}(f_j(z) \notin [L(j, z, t), U(j, z, t)]) \leq \frac{2\sqrt{2\log(T)}T|\mathcal{Z}|K}{T} \leq 2K|\mathcal{Z}|\sqrt{2\log(T)}.$$

Then, for the first term, by the same argument as [25],

$$\sum_{t=t_0}^{T} \mathbb{E}[U(J_t, Z_{J_t,t}, t) - L(J_t, Z_{J_t,t}, t)] \leq \sum_{t=t_0}^{T}\sum_{j=1}^{K}\sum_{z \in \mathcal{Z}}\mathbb{E}[U(j, z, t) - L(j, z, t)\mathbb{I}\{J_t = j, Z_{J_t,t} = z\}]$$

$$\leq 2\sqrt{\sigma^2(2 + 6\log(T))}\sum_{t=t_0}^{T}\sum_{j=1}^{K}\sum_{z \in \mathcal{Z}}\mathbb{E}\left[\frac{1}{\sqrt{2N_{j,z}(t)}}\mathbb{I}\{J_t = j, Z_{J_t,t} = z\}\right]$$

$$\leq 2\sqrt{\sigma^2(2 + 6\log(T))}\sum_{j=1}^{K}\sum_{z \in \mathcal{Z}}\mathbb{E}\left[\sum_{l=0}^{N_{j,z}(T)-1}\frac{1}{\sqrt{l+1}}\right]$$

$$\leq 2\sqrt{\sigma^2(2 + 6\log(T))}\sum_{j=1}^{K}\sum_{z \in \mathcal{Z}}\mathbb{E}\left[\sqrt{N_{j,z}(T)}\right]$$

$$\leq 2\sqrt{\sigma^2(2 + 6\log(T))}\sqrt{K|\mathcal{Z}|T}$$

where the last line follows by Cauchy-Schwartz. This concludes the proof. $\qquad\square$

# G    Further Experimental Results

## G.1    Posterior Distributions and Covariates

### G.1.1    $d$RGP-UCB

In this section, we plot the posterior (blue) of $d$RGP-UCB. with density given by the blue region in the instantaneous case. The red curve is the true recovery curve and the crosses are our observed samples for various values of $d$ and different kernels. Note that as the kernel gets smoother, the algorithm places more samples in the good regions. This is to be expected as for smoother kernels, there is less need to explore as many sub-optimal regions. Also, as $d$ increases more samples are at the peaks of the recovery curves.

(a) $d = 1$

(b) $d = 2$

(c) $d = 3$

Figure 5: $d$RGP-UCB with squared exponential kernel with $l = 0.5$

(a) $d = 1$

(b) $d = 2$

(c) $d = 3$

Figure 6: $d$RGP-UCB with squared exponential kernel with $l = 2$

(a) $d = 1$

(b) $d = 2$

(c) $d = 3$

Figure 7: $d$RGP-UCB with squared exponential kernel with $l = 5$

### G.1.2 *d*RGP-TS

In this section, we plot the posterior (blue) of *d*RGP-TS. with density given by the blue region with different $l$'s and $d$'s. We see much the same pattern as for *d*RGP-UCB, although it does seem to demonstrate poorer estimation of the recovery curve in the single step case. This suggests that the Thompson sampling approach is focusing on exploitation rather than exploration, as has been observed in other settings (eg. in linear bandits [1] show that the variance of the posterior needs to be inflated to encourage more exploration in Thompson Sampling). However, it is worth noting that the algorithms have only been run once for these plots.

(a) $d = 1$

(b) $d = 2$

(c) $d = 3$

Figure 8: *d*RGP-TS for squared exponential kernel with $l = 0.5$

(a) $d = 1$

(b) $d = 2$

(c) $d = 3$

Figure 9: $d$RGP-TS for squared exponential kernel with $l = 2$

(a) $d = 1$

(b) $d = 2$

(c) $d = 3$

Figure 10: $d$RGP-TS wit squared exponential kernel with $l = 5$

## G.2 Implementation of RogueUCB-Tuned

We briefly discuss the steps that were taken to map the recovering bandits problem into the setup of [18]. For this, we need to encode the recovery dynamics into a state dynamics function used by [18]. This can trivially be done by defining the functions $h : (\mathcal{Z}, \mathcal{A}) \rightarrow \mathcal{Z}$ as $h(\mathbf{z}, j) = \max\{\mathbf{z} + \mathbf{1}, z_{\max}\} - \max\{z_j + 1, z_{\max}\}\mathbf{e}_j$, where $\mathbf{1}$ is the vector of ones, $\mathbf{e}_j$ is the standard basis vector consisting of all zeros and a 1 in position $j$, and the maximum is taken component wise. As in [18], we did not implement the RogueUCB algorithm, but rather the empirical version, RogueUCB-Tuned, for which there are no theoretical guarantees. When implementing this, we set the parameter $\eta$ to be the maximal value of the KL-divergence, as in [18].

## G.3 Values of Theta used in Parametric Experiments

Here we give the values of $\theta$ (to 3dp) which were used in the logistic and gamma experiments in Section 7. These were sampled uniformly. Note that this sampling had no influence over our choice of kernel.

### G.3.1 Logistic

Table 2: $\theta$ values used in experiments with logistic recovery functions

|  | $\theta$ | | |
|---|---|---|---|
| Arm 1 | 0.584 | 0.521 | 12.239 |
| Arm 2 | 0.971 | 0.357 | 10.460 |
| Arm 3 | 0.121 | 0.622 | 25.631 |
| Arm 4 | 0.240 | 0.943 | 18.870 |
| Arm 5 | 0.613 | 0.925 | 20.310 |
| Arm 6 | 0.480 | 0.914 | 1.452 |
| Arm 7 | 0.974 | 0.484 | 10.128 |
| Arm 8 | 0.780 | 0.422 | 0.396 |
| Arm 9 | 0.658 | 0.591 | 23.264 |
| Arm 10 | 0.687 | 0.753 | 7.908 |

### G.3.2 Gamma

Table 3: $\theta$ values used in experiments with gamma recovery functions

|  | $\theta$ | | |
|---|---|---|---|
| Arm 1 | 2.068 | 0.249 | 0.508 |
| Arm 2 | 5.023 | 0.375 | 0.551 |
| Arm 3 | 3.657 | 0.470 | 0.772 |
| Arm 4 | 0.560 | 0.176 | 0.569 |
| Arm 5 | 3.901 | 0.747 | 0.500 |
| Arm 6 | 0.600 | 0.145 | 0.266 |
| Arm 7 | 6.482 | 0.522 | 0.554 |
| Arm 8 | 13.645 | 0.748 | 0.678 |
| Arm 9 | 7.365 | 0.562 | 0.288 |
| Arm 10 | 2.705 | 0.593 | 0.381 |

## G.4 Results for Different Lengthscales

In this section, we present results for the parametric setting where we have used different lenghtscales for the kernel of the Gaussian process in our methods. The parametric functions that we are considering are quite smooth so we choose a squared exponential kernel and used $l = 5$ in the

Table 4: Total reward at $T = 1000$ for $l = 2.5$

| Setting | 1RGP-UCB ($l = 2.5$) | 1RGP-TS ($l = 2.5$) | RogueUCB-Tuned | UCB-Z |
|---|---|---|---|---|
| Logistic | 448.6 (441.1,456.6) | 452.5 (443.7,460.3) | 446.2 (438.2,453.5) | 242.6 (229.6,256.0) |
| Gamma | 145.1 (138.5, 151.5) | 155.8 (148.8,162.5) | 132.7 (111.0,144.5) | 116.8 (108.4,125.5) |

Table 5: Total reward at $T = 1000$ for $l = 7.5$

| Setting | 1RGP-UCB ($l = 7.5$) | 1RGP-TS ($l = 7.5$) | RogueUCB-Tuned | UCB-Z |
|---|---|---|---|---|
| Logistic | 465.1 (457.3,472.9) | 465.1 (457.4,472.7) | 446.2 (438.2,453.5) | 242.6 (229.6,256.0) |
| Gamma | 145.2 (139.8, 151.0) | 155.8 (149.0,162.5) | 132.7 (111.0,144.5) | 116.8 (108.4,125.5) |

main text, and present results here for $l = 2.5$ and $l = 7.5$. Note that in this setting looking at the smoothness of the recovery functions to inform a decision about the lengthscale is reasonable since we are comparing our algorithms to RogueUCB-Tuned of [18] which requires knowledge of the parametric family and Lipschitz constant of the recovery function.

The results for $l = 2.5$ are shown in Table 4 and Figure 11. The results for $l = 7.5$ are in Table 5 and Figure 12. From these results, we can see that in the Gamma case, our algorithms are almost invariant to the choice of $l$, obtaining similar results for all choices of $l$. In particular, for all three choices of $l$ considered, our algorithms considerably outperform RogueUCB-Tuned of [18]. In the logistic setting, there is slightly more variation in the performance of our algorithms when the lengthscale changes, although the results are still fairly similar. In this case, we see that choosing $l = 7.5$ leads to the best results for both of our algorithms. This is most likely due to the fact that logistic functions are quite smooth and $l = 7.5$ represents the smoothest GPs we have considered.

(a) Logistic setup, $l = 2.5$

(b) Gamma setup, $l = 2.5$

Figure 11: Cumulative instantaneous regret $l = 2.5$

(a) Logistic setup, $l = 7.5$

(b) Gamma setup, $l = 7.5$

Figure 12: Cumulative instantaneous regret $l = 7.5$