[Reviews · NeurIPS 2019]

Reviewer 1



The major point that needs to be clarified for me is the distinction between single play regret and multiple play regret. - first, regarding the d-lookahead regret, it seems E[R_T^{(d)}] is only defined for policies of a very particular form, that would select every d time step a leaf of the d-lookahead tree. Am I right? The UCB-Z algorithm is not of this type for example. It is a bit weird to have regret notion that only apply to very specific algorithms - second, regarding the distinction between single and multiple play regret, am I right that E[R_T^{(d,m)}]=E[R_T^{(d)}]? That is, you would in principle care about the multiple play lookahead regret, as one should be allowed to be play an arm multiple times during the d-time step on which we optimize. - then, if I understood correctly, I just don't see the point of single-play regret. From my understanding, it would be defined only for strategies which selects each arm at most once during d steps (and therefore d cannot be larger than K in this case, right?). How do you motivate such a restriction? Currently in the paper, the motivation is mostly "easier to analyze" and I think a practical motivation is also needed. When thinking of the concrete application of the proposed algorithms for example in a recommendation settings, some questions arise. First, which kernel would be chosen? For each arm, the recovery function is defined on the discrete space [0,...,z_\max]. Typical kernel k(x,x') depend on the euclidian distance between x and x', in that case k(z,z') simply depends on the integer |z-z'|? More importantly, z_\max seems to represent a "characteristic time" after which the mean of an arm doesn't evolve any more. Does it make sense to know the maximum characteristic time of all products? In practice, what type of time scales / values of z_\max would be chosen in a recommendation context? Some other remarks: - full-horizon regret: the comparison is with respect to an oracle to would act optimally for maximizing rewards up to horizon T in an MDP. This oracle is called "stationary deterministic", but I don't agree that it is stationary. The optimal solution in finite-horizon MDP is solution to some dynamic programming equations which in this case reduce to a finite backwards induction. As a consequence, the optimal policy depends on the remaining horizon, and is therefore not stationary. - the analysis is in terms of Bayesian regret assuming recovery functions are drawn from the GP used in the algorithms. It seems the experimental protocol leading to Figure 4 is different: the recovery functions have a parametric forms and the parameters are drawn uniformly: is it equivalent from sampling the function from some GP? what would be the kernel? If not, it is worth mentioning the discrepancy between the kernel used by the algorithms and the "kernel" from which the functions are drawn - Figure 2 and Table 1 are a bit far from where they are needed (in Section 7): would it be possible to re-group all figures on the same page? - the bibliography contains several references to arxiv preprints that have been published by then, e.g. [10,26,28]. Could you please correct that?

Reviewer 2



In this paper, the authors study the recovering bandit problem, where the expected reward of arms varies according to some unknown function of the elapsed time since it was played. This is a significant problem, which occurs in practice for recommendation or advertising, where to vary the proposed item or ad during time is relevant. This problem is very challenging from an algorithmic point of view, since in addition of handling a multi-armed bandit problem, one must learn the recovery functions. Moreover, as the rewards of arms change during time, maximizing the instantaneous regret is not optimal. Here, the authors propose to maximize the rewards of the best sequence of actions. Three definition of the regret are proposed: the instantaneous regret, the full horizon regret, and the d-step Lookahead Regret. Instead choosing the full horizon regret, which compete against the best deterministic policy which knows the recovery functions, the authors focus on the d-step Lookhead Regret, which leads to tractable algorithms. Proposition 1 states that any algorithm with a low d-step Lookhead Regret can also have a low full horizon regret when d is not too small. For estimating the recovery function the authors use Gaussian Process. The authors propose two algorithms: a fully Bayesian approach which uses Thompson Sampling to sample the leaf of the lookhead tree (a sequence of d arms equipped with d recovery functions), and a hybrid approach which uses a UCB like algorithm to select the best leaf. The main drawback of the d-step lookhead strategy is that this approach does not scale with d. To improve the computational efficiency, the authors propose to use optimistic planning in section 6. The algorithms are analyzed in the case of single play lookhead, where an arm can be played only once during d steps, and multiple play lookhead. In the case of single play lookhead, the upper bound of regret are in O(sqrt (KT \gamma_T log (TK|Z|)), that is surprising since it does not depend on the depth of the lookhead d. In fact, \gamma_T has a strong dependence in d. For instance for squared exponential kernels \gamma_T = O((log T)^(d+1). The reviewer thinks that the dependence in d of \gamma should be explicit. Moreover, the upper bound of the instantaneous regret is the same than the upper bound of d-step lookhead regret (Theorem 2,3 vs Corollary 7). This suggests that the bounds in Corollary 7 are not tight, or that the dependence in d is not the same. Could the authors clarify this point? The experiments done on synthetic problems show the efficiency of the approach. Overall, this is an interesting paper on a significant and challenging problem. There are some shortcomings or ways of improvement: - The setting is so general that it could be interesting to position it with respect to reinforcement learning with discounted rewards. - The dependence in d of the regret bounds should be clarified. ____________________________________________________________________ I read the rebuttal. Thanks for answering my concern. I raised my score. I recommend acceptation.

Reviewer 3



Summary and Contributions From a practical perspective, the problem studied here is interesting. The authors tackle the problem of recommendations for a special kind of items that are associated with phases in a user's purchasing behaviour (i.e. items that users only buy rarely, like a TV, a new phone, furniture, movies etc.). Here, a play of an arm triggers a state change for a period of time (users most likely do not want to watch the same movie again immediately). Mapping this to a bandit setting, the authors aim to solve the a MAB problem where, after each play of an arm j, the expected reward of j start changing over time according to an unknown function f_j drawn from a GP of known mean and kernel, until a maximum time z_max, after which the expected reward remains constant. The authors propose using d-lookahead regret as a proxy for the classical measure of regret (which is prohibitively expensive to compute in this setting) and show that it is indeed a good approximation (the approximation is arbitrarily close as T\to\infty). Additionally, the paper introduces the d-lookahead UCB and Thompson Sampling variants and analyse their regret proving scaling of the order O(sqrt(KT\gamma \log(T))) where \gamma depends on the GP kernel and takes values from O(log(T)) to O(T^{1/(1+\nu)}log(T)) for Matern(\nu) kernels. Overall I find this paper interesting but wonder whether some of the assumptions hurt the practical significance of the work. The GP kernel will not be known in most real-life scenarios - would the uninformative prior be sufficient? z_max being the same for all arms seems a bit unrealistic. Would it be possible to relax this assumption? Another questionable assumption that I think also needlessly complicates the problem is the fact that the decision maker is forced to play an arm each round (normally, web services are not forced to display ads, for example). Having the ability to pass on playing at some round would allow for easier planning ahead since now, there is more independence between arms. While I have not checked the proofs, in my opinion the paper is very well written and I particularly like the result on d-lookahead regret being a good approximation of the full horizon regret. I like this metric and think it's a great proxy for this setting. I appreciate the authors comparing their algorithm's performance to several baselines, including the common sense UCB-Z algorithm. The authors also provide a section on reducing the computational complexity of their algorithm. Despite some questionable assumptions, I can see this paper representing a solid starting point for further work on this setting, I therefore recommend accepting this paper. =========== Post Rebuttal =========== I have read the author's reply and am satisfied with the clarifications.

[Author Response · NeurIPS 2019]

# Author Response: Recovering Bandits #7889

We would like to thank all the reviewers for their detailed and constructive comments. Several reviewers pointed out that the paper would benefit from more discussion of the applicability of our algorithms in recommendation systems. We propose to add a paragraph to the end discussing these practical challenges. We now address each reviewer in turn. All references correspond to the bibliography of the submitted paper.

Reviewer 1:
- The $d$-step lookahead regret it is the regret in batches of size $d$ and so it is valid for any sort of policy. We can measure the d-step lookahead regret for UCB-Z. However, it may be high since UCB-Z is a greedy strategy which does not look at future states so may play arms which lead to bad reward in the remaining steps in the lookahead.
- We agree that the multiple play regret is more practically relevant, and focused on this in the optimistic planning extension and experiments. However, we believe that first studying the single play regret allows us to gain more insights into the problem. Particularly, we show that in the single play case we can achieve the same rate of regret as the instantaneous case, whereas in the multiple play case we are penalized for not updating the posterior between repeated plays of the same arm. This allows us to conclude that it is the lack of updating that makes the multiple play lookahead case more difficult. We will clarify the benefit of studying the single play case in the final version.
- If we are minimizing regret to some fixed horizon, the optimal dynamic programming solution would indeed depend on this horizon. However, we are interested in anytime algorithms, which do not depend on the horizon. In this case, the oracle is stationary.
- A typical choice of kernel depends on $|z - z'|$ (e.g. Gaussian kernel). By scaling this distance we can interpolate between the case where the rewards are completely unrelated for different $z$ values, and the case with essentially constant reward. Prior knowledge of the smoothness of the functions can be used to tune this scaling factor. Alternatively, it may be possible to adaptively tune it (see e.g. Chapter 5 of [24]). We will discuss this in the additional paragraph.
- For the choice of $z_{\max}$, we first point out that we only require an upper bound on it. We believe that in practice, this is reasonable. As an example for the scale, if we suggest one item per day, the reward from a user not having seen it for 30 days is likely to be the same as if they have not seen it for 60 days, so $z_{\max} = 30$, and 60 is an upper bound. Adaptively choosing $z_{\max}$ is an interesting area for future work. The choice of $z_{\max}$ will be discussed in the extra paragraph.
- In Figure 4, the parameters were sampled uniformly as a method for selecting the parameters of the 'true' recovery curves. We did not use this to influence our choice of kernels for our algorithms. This will be clarified.
- We will correct the issue with some of the references and figures not appearing in the correct place.

Reviewer 2:
- We believe there has been some confusion regarding the definition of $\gamma_T$. You are correct that in [28], bounds on $\gamma_T$ of the form $O(\log(T)^{D+1})$ are given for the squared exponential kernel where D is the dimension of the input space. When we apply this result, our input space is the set of integers $\{0, z_{\max}\}$ which is one dimensional. Hence, $\gamma_T = O(\log(T)^2)$. There is no dependence on $d$, the depth of the lookahead. We apologize if our notation caused confusion, and will clarify this in the final version.
- The single play d-step lookahead regret of our algorithms are of the same order as the instantaneous regret. At first this may appear surprising, however, it can be explained by noting that in the single play case, we select a sequence of d arms at (approximately) $T/d$ time steps, and since we can only play an arm at most once in a d-step lookahead, the posterior can be updated after each play meaning that we do not lose any information. This will be clarified.
- You are correct that our problem is related to reinforcement learning (as discussed in lines 144–145). However, even if posed as such, it is still challenging since there are no discount factors and the states are never reset.

Reviewer 3:
- Our model captures a wide range of recovery functions, ranging from uncorrelated $f_j(z)$'s to more complex functions. Under mild assumptions most of these prior kernels will be sufficient to represent the true function (if this is relatively smooth). However, we learn faster with good priors, so we suggest selecting the prior based on the smoothness of the functions, which may be known in advance. We will discuss this, and adaptive methods, in the additional paragraph.
- Our approach can be used in the setting with arm-dependent $z_{\max}$ as long as there is an upper bound that holds for all arms. Indeed, we can extend $f_j$ from the arm-dependent value, $z_{\max,j}$, to $z_{\max}$ using the fact that $f_j(z) = f_j(z_{\max,j})$ for $z = z_{\max,j} + 1, \ldots, z_{\max}$ (e.g. in the experiment with logistic recovery curves, in effect $z_{\max,j}$ was arm-dependent so $f_j$ was constant above that value). We will add a discussion of this in the extra paragraph. An interesting area for further work is to develop more principled methods for estimating the $z_{\max,j}$ of each arm within the algorithm.
- There are two ways to formulate skipping rounds. In the first, rounds can be skipped at no cost to the regret. However this may not be practically relevant. Alternatively, skipping a round can incur some regret. In this case, our algorithms can be applied with an additional 'pseudo-arm' with constant negative reward which is played when a round is skipped.
- To map recovering bandits to the setup of [18], we need to define state dynamics matrices. In the notation of [18], $A$ sends $z$ to $z + 1$, and $B$ is $-\infty$, so that the projection onto $[0, z_{\max}]$ is 0. We implemented RogueUCB-Tuned with the parameter $\eta$ set to the maximal KL-divergence, as in [18]. We will specify this in the final version.

[Meta-Review · NeurIPS 2019]

A solid contribution on a specific bandit model, strongly defended by all reviewers (and me).